# A trimeric *Cr*RLK1L-LLG1 complex genetically modulates SUMM2-mediated autoimmunity

Yanyan Huang [1,2,6], Chuanchun Yin[1,3,6], Jun Liu[2,6], Baomin Feng[2,4], Dongdong Ge[3], Liang Kong[2], Fausto Andres Ortiz-Morea[2], Julia Richter[5], Marie-Theres Hauser [5], Wen-Ming Wang[1], Libo Shan[3] & Ping He [2✉]

Cell death is intrinsically linked with immunity. Disruption of an immune-activated MAPK cascade, consisting of MEKK1, MKK1/2, and MPK4, triggers cell death and autoimmunity through the nucleotide-binding leucine-rich repeat (NLR) protein SUMM2 and the MAPK kinase kinase MEKK2. In this study, we identify a *Catharanthus roseus* receptor-like kinase 1-like (*Cr*RLK1L), named LETUM2/MEDOS1 (LET2/MDS1), and the glycosylphosphatidylinositol (GPI)-anchored protein LLG1 as regulators of *mekk1-mkk1/2-mpk4* cell death. LET2/MDS1 functions additively with LET1, another *Cr*RLK1L, and acts genetically downstream of MEKK2 in regulating SUMM2 activation. LET2/MDS1 complexes with LET1 and promotes LET1 phosphorylation, revealing an intertwined regulation between different *Cr*RLK1Ls. LLG1 interacts with the ectodomain of LET1/2 and mediates LET1/2 transport to the plasma membrane, corroborating its function as a co-receptor of LET1/2 in the *mekk1-mkk1/2-mpk4* cell death pathway. Thus, our data suggest that a trimeric complex consisting of two *Cr*RLK1Ls LET1, LET2/MDS1, and a GPI-anchored protein LLG1 that regulates the activation of NLR SUMM2 for initiating cell death and autoimmunity.

---

[1] State Key Laboratory of Crop Gene Exploration and Utilization in Southwest China, Sichuan Agricultural University, 611130 Chengdu, P. R. China. [2] Department of Biochemistry & Biophysics, Institute for Plant Genomics & Biotechnology, Texas A&M University, College Station, TX 77843, USA. [3] Department of Plant Pathology & Microbiology, Institute for Plant Genomics & Biotechnology, Texas A&M University, College Station, TX 77843, USA. [4] State Key Laboratory of Ecological Control of Fujian-Taiwan Crop Pests, Key Laboratory of Ministry of Education for Genetics, Breeding and Multiple Utilization of Crops, Plant Immunity Center, Fujian Agriculture and Forestry University, 350002 Fuzhou, P. R. China. [5] Department of Applied Genetics and Cell Biology, University of Natural Resources and Life Sciences, Vienna (BOKU), 18 A-1190 Muthgasse, Austria. [6] These authors contributed equally: Yanyan Huang, Chuanchun Yin, Jun Liu. ✉email: pinghe@tamu.edu

Being sessile and lacking the adaptive immunity, plants have evolved two-tiered immune receptors to detect infections. The plasma membrane-associated immune receptors, termed pattern-recognition receptors (PRRs), sense pathogen- or microbe-associated molecular patterns (PAMPs or MAMPs), or host-derived danger-associated molecular patterns (DAMPs) that trigger immune responses against a broad spectrum of pathogens, including non-adapted pathogens[1–3]. PRRs are often receptor-like kinases (RLKs) and receptor-like proteins (RLPs) in plants[4,5]. The intracellular immune receptors, which are often nucleotide-binding domain leucine-rich repeat proteins (NLRs), recognize directly or indirectly pathogen-delivered effectors and trigger race-specific resistance against adapted pathogens carrying the cognate effectors[6–8]. The NLR-mediated immune response is usually associated with a rapid and localized cell death at the infection site, known as the hypersensitive response (HR), to restrict pathogen spread.

Plants have evolved a largely expanded number of RLKs[9]. The most well-studied RLKs contain an extracellular leucine-rich repeat (LRR) domain, called LRR-RLKs, which play important roles not only in regulating plant immunity by sensing MAMPs/DAMPs, but also in modulating plant growth and development by perceiving endogenous signals or environmental cues[10,11]. RLKs with an extracellular malectin-like domain, also called *Catharanthus roseus* RLK1-like kinases (*Cr*RLK1Ls), have long been known to be key regulators in various developmental processes including cell elongation, polarized growth, and fertilization[12–15]. Among 17 members in *Arabidopsis*, FERONIA (FER) is involved in a myriad of biological processes including fertilization, root hair growth, plant hormone signaling, and immunity[16–18]. ANXUR1 (ANX1) and ANX2, close homologs of FER, play redundant roles in cell wall integrity during pollen tube growth[19–21]. BUDDHA'S PAPER SEAL 1 (BUPS1) and BUPS2 interact with ANX1/ANX2 in maintaining pollen tube integrity[22,23]. In addition, both FER and ANXs are involved in plant immunity[24–27]. FER scaffolds MAMP-induced PRR complex formation[24] and suppresses jasmonic acid hormone signaling in plant immunity[26]. However, ANX1 and ANX2 negatively regulate two-tiered plant immunity by modulating both PRRs and NLRs[25]. The glycosylphosphatidylinositol (GPI)-anchored protein LORELEI (LRE) and LRE-like proteins LLGs function as co-receptors/adapters for FER in regulating plant growth, reproduction and immunity[28,29]. Recently, it has been shown that LLG2/LLG3 are co-receptors of BUPSs/ANXs in regulating pollen tube integrity[30,31]. Interestingly, LLG1 is also involved in plant immunity by association and modulation of PRR FLS2[32].

Although the recognition of pathogens by the innate immune system differs, common responses and signaling components converge at multiple levels[1,2,6]. The mitogen-activated protein kinase (MAPK or MAP kinase) cascades are among essential modules regulating both PRR and NLR-mediated immune responses in plants[33–35]. The classical MAPK cascade consists of three sequentially phosphorylated kinases, including MAPK kinase kinases (MAPKKKs, MKKKs, or MEKKs), MAPK kinases (MAPKKs, or MKKs), and MAPKs (MPKs)[36]. Two parallel MAPK cascades, MKKK3/5-MKK4/5-MPK3/6 and MEKK1-MKK1/2-MPK4, play important roles in PRR signaling[37–39]. Plants with deficiency in MEKK1, MKK1/2 or MPK4 display autoimmune phenotypes and are seedling lethal[40–44]. The autoimmunity in *mekk1*, *mkk1/2*, and *mpk4* mutants is due to the activation of the NLR protein SUPPRESSOR OF *mkk1 mkk2* (SUMM2)-mediated defense[45]. Intriguingly, the MEKK1-MKK1/2-MPK4 cascade negatively regulates another MAPKKK MEKK2, which interacts with MPK4, and positively regulates the NLR SUMM2-triggered autoimmunity[46,47]. Furthermore, the transcript and protein abundance of MEKK2 is positively correlated

with its ability to trigger autoimmunity[47]. Another kinase, CALMODULIN-BINDING RECEPTOR-LIKE CYTOPLASMIC KINASE 3 (CRCK3), which is phosphorylated by MPK4, is also required for SUMM2-activated autoimmunity[48]. Apparently, a PRR-activated MAPK cascade, consisting of MEKK1-MKK1/2-MPK4, functions genetically upstream of SUMM2 in regulating autoimmunity.

To gain insights into the mechanisms underlying SUMM2-mediated defense, which is otherwise suppressed by a PRR-activated MAPK cascade, we deployed a transient RNAi-based genetic screen by virus-induced gene silencing (VIGS) and screened for suppressors of *mekk1* cell death from a collection of T-DNA insertion mutants. We identified *lethality suppressor of mekk1* 1 (*letum1* or *let1*) that largely suppressed the autoimmunity in *mekk1*, *mkk1/2*, and *mpk4*. LET1 is a member of uncharacterized *Cr*RLK1Ls[49]. In this study, we have screened additional *Cr*RLK1Ls and identified LET2/MEDOS1 (MDS1) in regulating *mekk1*, *mkk1/2*, and *mpk4* autoimmunity. Both *let1* and *let2* single mutants suppressed *mekk1*, *mkk1/2*, and *mpk4* cell death, however, the *let1let2* (called *let1/2* henceforth) double mutant showed further suppression, indicating the additive function of LET1 and LET2/MDS1 in modulating SUMM2 activation. Interestingly, LET1 and LET2/MDS1 heteromerize, and LET2/MDS1 promotes LET1 phosphorylation, suggesting a phosphoregulation between different *Cr*RLK1Ls. Similar to LET1 and LET2/MDS1, the GPI-anchored protein LLG1, but not LLG2, LLG3, nor LRE, plays a role in modulating *mekk1*, *mkk1/2*, and *mpk4* cell death, and functions genetically downstream of MEKK2 and upstream of SUMM2. Likely as a co-receptor, LLG1 interacts with the ectodomain of LET1 and likely LET2/MDS1 and mediates LET1/2 transport to the plasma membrane. Thus, our results suggest that a specific trimeric *Cr*RLK1L module consisting of LET1, LET2/MDS1, and the GPI-anchored LLG1 modulates SUMM2-mediated autoimmunity.

## Results

**The mutations in the *CrRLK1L* gene, *LET2/MDS1*, suppress RNAi-*MEKK1* cell death.** There are 17 *CrRLK1L* genes in the *Arabidopsis* genome (Fig. 1a). Among them, *LET1* (*AT2G23200*) was identified as a modulator of autoimmunity in *mekk1*, *mkk1/2*, and *mpk4*[49]. To systematically investigate the *CrRLK1L* gene family members in this process, we collected the T-DNA insertion lines of individual *CrRLK1L* genes and determined their roles on silencing *MEKK1*-triggered cell death through a VIGS approach (Supplementary Fig. 1a). Among 20 T-DNA insertion lines, including the *herk1-1the1-4* double mutant, five lines of three genes, *AT5G24010* (two lines), *AT4G39110* (*BUPS1*, two lines), and *AT2G21480* (*BUPS2*, one line), do not bear T-DNA insertions in the annotated sites and were characterized as wild type (WT; Supplementary Fig. 1a). It has been shown that the *bups* mutants have defects in pollen tube growth[22]. The remaining 15 T-DNA insertion lines are homozygous mutants (Supplementary Fig. 1a). Among them, two mutants, *SALK_139579* and *SALK_066322*, but not the other 13 mutants of 12 *CrRLK1Ls*, suppressed the growth defects and cell death caused by RNAi-*MEKK1* (Fig. 1b, c and Supplementary Fig. 1b). *SALK_139579* bears a T-DNA insertion in the signal peptide (SP) motif, and *SALK_066322* has a T-DNA insertion in the malectin-like domain of *AT5G38990*, respectively (Fig. 1b and Supplementary Fig. 1a). Since they suppressed RNAi-*MEKK1*-mediated cell death, *AT5G38990* was named as *LET2*, and the corresponding mutants *SALK_139579* and *SALK_066322* were named as *let2-1* and *let2-2*.

*LET2* has been previously named as *MEDOS1* (*MDS1*) and is involved in growth responses to metal ions[50]. Notably, *LET2/*

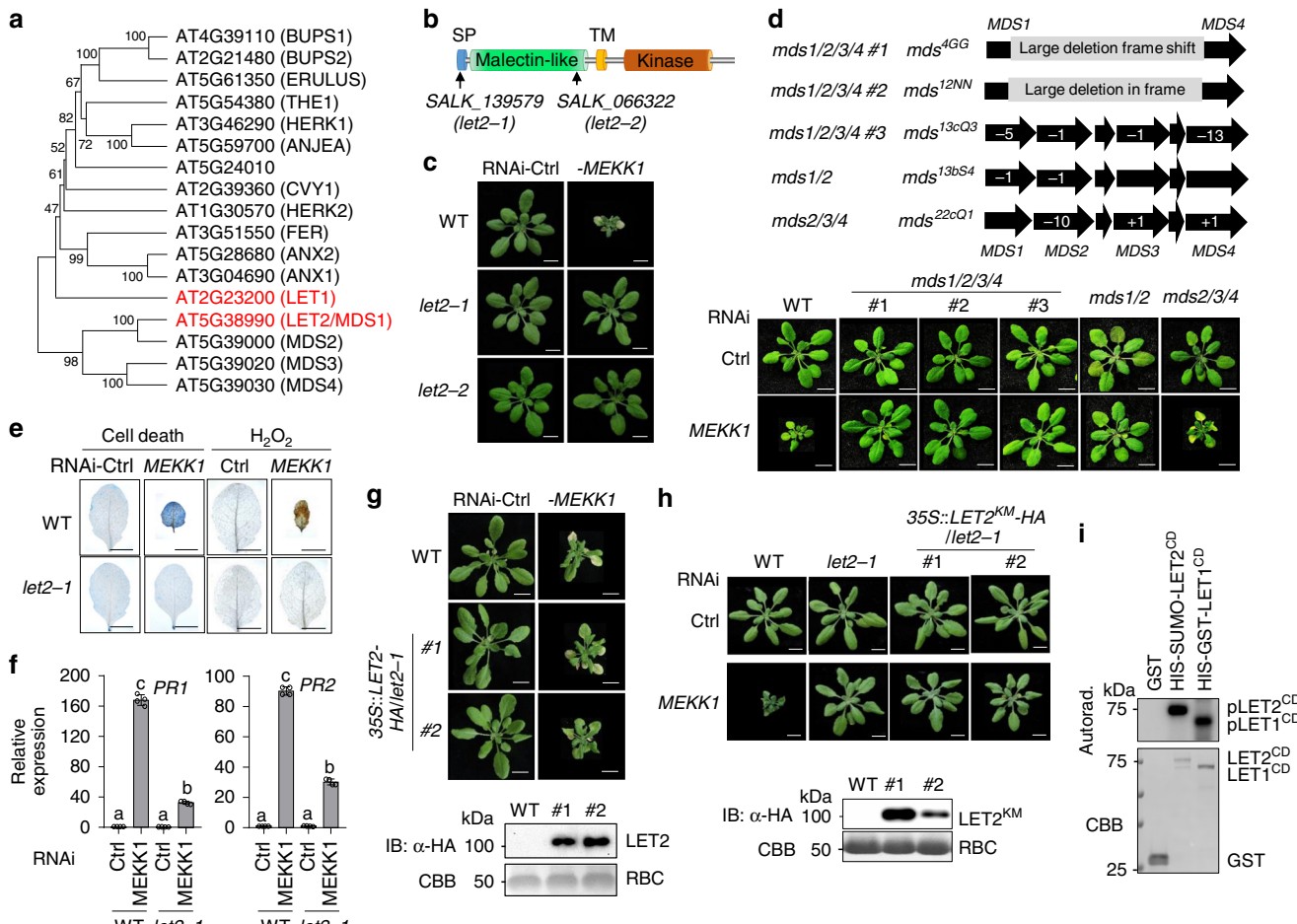

**Fig. 1 LET2/MDS1 is involved in RNAi-MEKK1 cell death. a** A phylogenetic tree of *Cr*RLK1L family proteins. The full-length protein sequences were used to generate the phylogenetic tree by the UPGMA method with 1000 bootstrap replicates in MEGA-X. **b** A schematics depicting LET2/MDS1 protein motifs and T-DNA insertion sites in the *let2* mutants. LET2/MDS1 consists of the N-terminal signal peptide (SP), a malectin-like domain, a transmembrane domain (TM), and a cytosolic kinase domain. The arrows indicate the T-DNA insertion sites of the indicated mutant alleles. **c** The *let2* mutants suppress plant dwarfism and leaf chlorosis induced by silencing *MEKK1*. The plant images were photographed at 3 weeks after inoculation with Agrobacterium carrying the indicated VIGS vectors. Ctrl is the vector containing GFP. Scale bar, 1 cm. **d** The *LET2/MDS1*, but not *MDS2*, *MDS3*, nor *MDS4*, is involved in RNAi-*MEKK1* cell death. The *mds1/2/3/4*, *mds1/2*, and *mds2/3/4* CRISPR/Cas knockout plants were inoculated with Agrobacterium carrying the indicated VIGS vectors. The plant images were photographed at 3 weeks after inoculation. The chromosome locations and Indels of *MDS1*, *MDS2*, *MDS3*, and *MDS4* on individual mutants are shown on the top. Scale bar, 1 cm. **e** The *let2-1* mutant suppresses cell death and $H_2O_2$ accumulation triggered by silencing *MEKK1*. The leaves were detached from plants in **c** and stained by trypan blue for cell death (left panel) and DAB for $H_2O_2$ accumulation (right panel). Scale bar, 0.5 cm. **f** The *let2-1* mutant suppresses the expression of *PR* genes triggered by silencing *MEKK1*. The expression of *PR1* and *PR2* from plants in **c** was normalized to the expression of *UBQ10* and the data are shown as the mean ± SE of four biological repeats ($n = 4$). $P = 3.00 \times 10^{-14}$ (*PR1*, column 1 and 2), $P = 1.60 \times 10^{-7}$ (*PR1*, column 3 and 4), $P = 4.40 \times 10^{-14}$ (*PR1*, column 2 and 4), $P = 3.00 \times 10^{-14}$ (*PR2*, column 1 and 2), $P = 3.45 \times 10^{-11}$ (*PR2*, column 3 and 4), and $P = 5.10 \times 10^{-14}$ (*PR2*, column 2 and 4). The different letters indicate the significant difference determined by one-way analysis of variance (ANOVA) followed by the Tukey test ($P < 0.05$). **g** Expression of *LET2/MDS1* in *let2-1* restores the cell death triggered by silencing *MEKK1*. The plant images were taken at 3 weeks after inoculation with Agrobacterium carrying the indicated VIGS vectors. #1 and #2 are two independent *35 S::LET2-HA* transgenic lines in *let2-1*. Scale bar, 1 cm. Protein expression of LET2-HA in transgenic lines is shown on the bottom. The total proteins were immunoblotted by an α-HA antibody (upper panel). Coomassie Brilliant Blue (CBB) staining of RuBisCO (RBC) is shown as a loading control (lower panel). The molecular weight (MW) was labeled on the left of immunoblots as kDa. **h** The LET2/MDS1 kinase mutant cannot complement *let2-1*. Two lines of transgenic plants carrying the kinase-inactive mutant of $LET2^{KM}$ (K554E) driven by a *35S* promoter in *let2-1* are shown. Scale bar, 1 cm. Protein expression of $LET2^{KM}$-HA in transgenic lines is shown on the bottom. **i** LET2/MDS1 bears kinase activity in vitro. GST and the LET2/MDS1 cytosolic kinase domain (HIS-SUMO-$LET2^{CD}$) proteins were purified from *E. coli*. The LET1 cytosolic kinase domain (HIS-GST-$LET1^{CD}$) proteins were purified from insect cells. The kinase assay was performed with [γ-$^{32}$P] ATP. CBB staining was used as a loading control. The above experiments were repeated three times with similar results.

*MDS1* belongs to the *MDS1-4* subfamily which resides in a tandem repeat region with three additional *Cr*RLK1Ls, *MDS2*, *MDS3*, and *MDS4* (Fig. 1a, d). *MDS3* and *MDS4* genes have redundant function in plant growth adaptation upon exposure to excess nickel ions[50]. We tested whether *MDS* genes also have redundant function in regulating RNAi-*MEKK1* cell death with the CRISPR/Cas9-generated double, triple, and quadruple *mds* mutants[50]. The *mds1/2/3/4* mutants #1 (*mds*4GG) and #2 (*mds*12NN) contain large deletions from *MDS1* to *MDS4*; the *mds1/2/3/4* #3 (*mds*13cQ3) contains deletions in four individual *MDS* genes; the *mds2/3/4* mutant (*mds*22cQ1) has Indels in *MDS2*, *MDS3*, and *MDS4*; and the *mds1/2* mutant (*mds*13bS4) has deletions in *MDS1* and *MDS2* (Fig. 1d)[50]. The individual mutants of *mds2*, *mds3*, and *mds4* did not affect RNAi-*MEKK1* cell death

(Supplementary Fig. 1b). Consistent with the role of *LET2/MDS1* in *MEKK1* cell death regulation, the *mds1/2/3/4* (#1, #2, #3), and *mds1/2* mutants, but not the *mds2/3/4* mutant, largely suppressed RNAi-*MEKK1* cell death (Fig. 1d). Thus, the data support that *LET2/MDS1* is a major *CrRLK1L* gene involved in the modulation of *mekk1* cell death.

The *let2-1* mutant suppressed RNAi-*MEKK1* cell death detected by trypan blue staining and $H_2O_2$ accumulation by 3,3′-diaminobenzidine (DAB) staining compared to WT plants (Fig. 1e). The *let2-1* mutant also suppressed the constitutive activation of defense marker genes, including *pathogenesis-related 1* (*PR1*) and *PR2*, caused by silencing *MEKK1* (Fig. 1f). To confirm that the causal mutation in *let2* is *AT5G38990*, we transformed the full-length cDNA of *AT5G38990* under the control of a *35S* promoter tagged with a double HA epitope at the carboxyl (C)-terminus (*35S::LET2-HA*) into the *let2-1* mutant. The *35S::LET2-HA* transgenic plants restored the RNAi-*MEKK1* cell death in *let2-1* (Fig. 1g). To determine whether the kinase activity of LET2/MDS1 is required for its function in the *mekk1* cell death pathway, we mutated a conserved lysine residue in the ATP-binding loop of LET2/MDS1 to glutamic acid (K554E, LET2 kinase-inactive mutant LET2$^{KM}$) and generated transgenic lines expressing *LET2$^{KM}$* in *let2-1*. Unlike *35S::LET2-HA/let2-1*, the *35S::LET2$^{KM}$-HA/let2-1* transgenic plants did not restore the cell death caused by silencing *MEKK1* (Fig. 1h), suggesting that its kinase activity is required for LET2/MDS1 function in *mekk1* cell death regulation. Consistently, the LET2/MDS1 cytosolic domain (CD) consisting of the juxtamembrane and kinase domains fused with HIS-tagged SUMO enzyme target peptide (HIS-SUMO-LET2$^{CD}$) displayed autophosphorylation activity in an in vitro kinase assay, similar with HIS-GST-LET1$^{CD}$, suggesting that LET2/MDS1 is an active kinase (Fig. 1i).

**LET1 and LET2/MDS1 function additively in modulating *mekk1*, *mkk1/2*, and *mpk4* cell death.** To genetically confirm the function of *LET2/MDS1* in *mekk1* cell death, we generated the *let2mekk1* double mutant by crossing the *let2-1* and *mekk1$^{+/-}$* (*mekk1* is heterozygous) mutants. The *let2mekk1* double mutant significantly alleviated the growth defects and dwarfism of *mekk1* when grown on ½MS plates (Fig. 2a, b). Since the *let1mekk1* double mutant also suppressed the growth defects of *mekk1*[49], we compared the phenotype of *let2mekk1* and *let1mekk1*. The *let2-mekk1* mutant was slightly smaller than *let1mekk1* at 2-week-old stage (Fig. 2a, b). At the reproductive stage when grown on soil, the *let2mekk1* mutant is obviously smaller than *let1mekk1*, displaying stronger cell death and failing to bolt (Fig. 2c). Interestingly, the *let1/2mekk1* triple mutant grew bigger and had more fresh weight than the *let1mekk1* and *let2mekk1* mutants at both seedling (Fig. 2a, b) and the reproductive (Fig. 2c) stages. The *let1/2mekk1* mutant normally bolted and produced seeds (Fig. 2c). These data indicate that *let2* suppresses cell death caused by either silencing or mutation of *MEKK1*, and *LET1* and *LET2/MDS1* function additively in modulating *mekk1* cell death.

The MEKK1 pathway is mediated through MKK1/2 and MPK4. Similar as *mekk1*, the *mkk1/2* double mutant and the *mpk4* mutant are seedling lethal[43,44]. We tested whether the *let2* mutant interferes with *mkk1/2* and *mpk4* cell death by generating the *let2mkk1/2* triple mutant, and the *let2mpk4* double mutant. The *let2mkk1/2* mutant largely alleviated *mkk1/2* cell death (Fig. 2d, e). The *let1/2mkk1/2* quadruple mutant grew better than *let2mkk1/2* and *let1mkk1/2* triple mutants, with fewer dead leaves at the 4-week-old stage (Fig. 2d, e). In addition, the *let2mpk4* double mutant suppressed *mpk4* cell death (Fig. 2f, g). The data indicate that LET2/MDS1 functions genetically downstream of MPK4 in the *mekk1-mkk1/2-mpk4* cell death pathway. The

*let2mpk4*, *let1mpk4*, and *let1/2mpk4* mutants were in the ascending order of plant size and fresh weight (Fig. 2f, g), corroborating the notion that LET2/MDS1 acts additively with LET1 in modulating the *mekk1-mkk1/2-mpk4* cell death.

*MEKK1* belongs to a tandemly duplicated gene family with *MEKK2* and *MEKK3*. The *mekk2* mutant suppressed *mekk1*, *mkk1/2*, and *mpk4* cell death[46,47]. Notably, the plant size and fresh weight of *let1/2mpk4* were similar with those of *mekk2mpk4* (Fig. 2f, g). To dissect whether LET1/2 and MEKK2 function independently or in a same pathway in regulating the *mekk1-mkk1/2-mpk4* cell death, we generated the *mekk2let1/2mpk4* quadruple mutant. The plant size and fresh weight of *mekk2let1/2mpk4* are not significantly different from those of *let1/2mpk4* or *mekk2mpk4* (Fig. 2f, g), suggesting that LET1 and LET2/MDS1 function genetically in the same pathway with MEKK2. The *mpk4* mutant displays the increased root width, which is independent of MEKK2[46]. Similarly, the increased root width in *mpk4* was not suppressed in the *let1mpk4*, *let2mpk4*, *let1/2mpk4*, or *mekk2let1/2mpk4* mutants (Supplementary Fig. 2), suggesting that LET1 and LET2/MDS1 are not involved in MPK4-regulated root development.

**LET2/MDS1 functions genetically downstream of MEKK2 and upstream of SUMM2.** Since both LET2/MDS1 and MEKK2 are required for SUMM2 activation, we tested the genetic relationship of LET2/MDS1 with MEKK2 and SUMM2. Overexpression of *MEKK2* under a constitutive *35S* promoter induced growth defects, cell death, $H_2O_2$ accumulation, and expression of *PR* genes in WT background, which were positively correlated to the MEKK2 protein level (Fig. 3a–d). We have obtained 75 independent transgenic plants carrying *35S::MEKK2-HA* at the $T_1$ generation with positive signals by α-HA immunoblots. We further classified them into four categories according to the growth defect severity: 16% (12 out of 75) plants exhibited severe dwarfism and cell death; 25.3% (19 out of 75) showed moderate dwarf and cell death; 26.7% (20 out of 75) exhibited further alleviated dwarfism with relatively big leaves and 32% (24 out of 75) exhibited weak dwarfism (Fig. 3a). We also generated 70 independent transgenic plants at the $T_1$ generation expressing *35S::MEKK2-HA* in the *let2-1* background with immunoblot positive signals for MEKK2-HA. Overall, the plant dwarfism and growth defects triggered by overexpressing *MEKK2* in WT were alleviated in *let2-1* with 4.3% (3 out of 70) of plants showing severe dwarfism and cell death, 7.1% (5 out of 70) showing moderate dwarfism, 34.3% (24 out of 70) showing weak dwarfism, and 54.3% (38 out of 70) showing slightly smaller size than *let2-1* (Fig. 3a). The cell death, $H_2O_2$ accumulation, and expression of *PR* genes caused by overexpressing *MEKK2* were also reduced in *let2-1* compared to WT plants (Fig. 3c, d). Notably, the protein expression level of MEKK2 was similar in *let2-1* and WT plants (Fig. 3b). The data indicate that LET2/MDS1 is required for overexpressing *MEKK2*-activated cell death and functions genetically downstream of MEKK2.

It has been reported that the active SUMM2 (SUMM2$^{ac}$), which bears an aspartate-to-valine mutation at the 478$^{th}$ amino acid residue in the methionine-histidine-aspartic acid (MHD) motif triggers cell death in *Nicotiana benthamiana*[45]. To delineate the genetic relationship of SUMM2 and LET2/MDS1 in cell death regulation, we generated *35S::SUMM2$^{ac}$-HA* transgenic plants in WT and *let2-1*. About 52.9% (36 out of 68) of *35S::SUMM2$^{ac}$-HA* transgenic plants in WT showed growth defects, cell death, $H_2O_2$ accumulation, and elevated expression of *PR* genes (Fig. 3e–h). The *35S::SUMM2$^{ac}$-HA* transgenic plants in *let2-1* showed a similar level of plant growth defects and dwarfism with 53.5% (38 out of 71) of plants (Fig. 3e–h). The protein expression level of

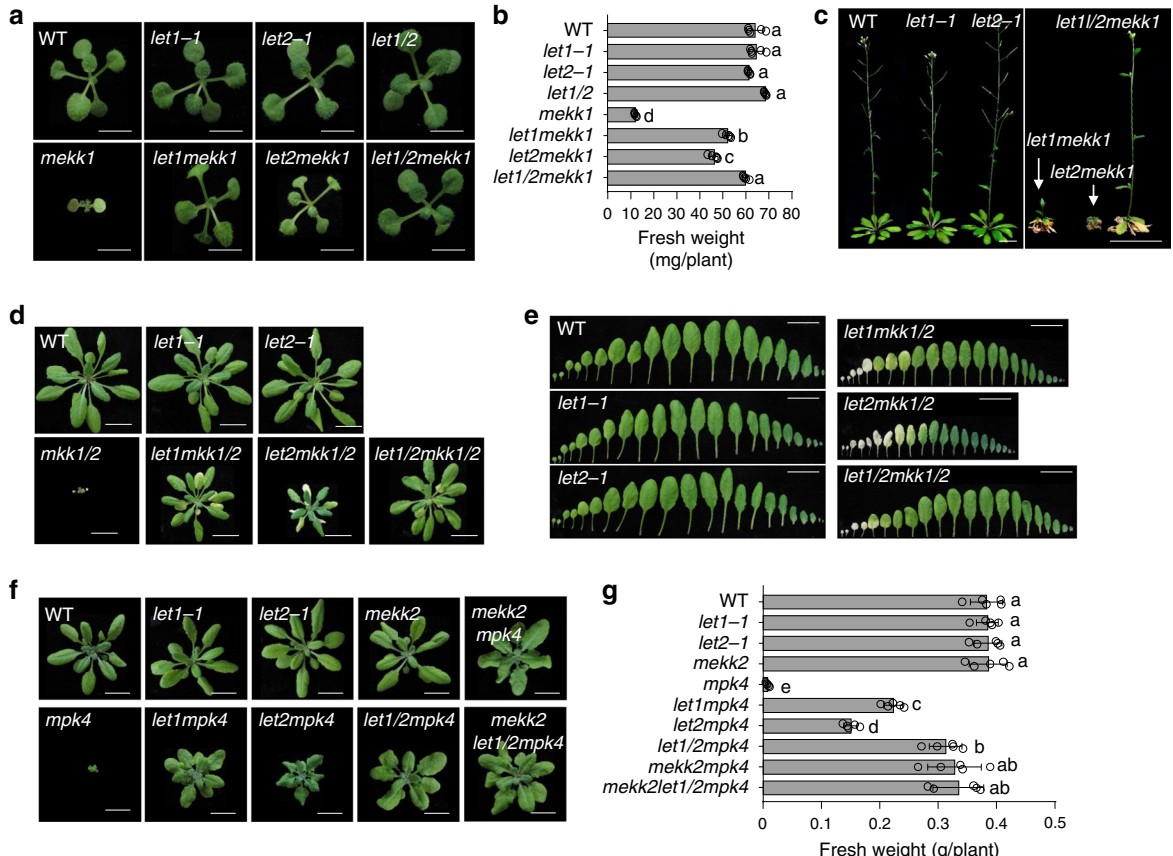

**Fig. 2 LET1 and LET2/MDS1 function additively in regulating *mekk1*, *mkk1/2*, and *mpk4* cell death. a–c** LET1 and LET2/MDS1 function additively in regulating *mekk1* cell death. **a** Two-week-old plants of different genotypes grown on ½MS plates are shown. Scale bar, 0.5 cm. **b** The fresh weight of the indicated plants in **a**. The data are shown as the mean ± SE ($n = 5$). $P = 1.00 \times 10^{-13}$ (column 5 and 6), $P = 1.00 \times 10^{-13}$ (column 5 and 7), $P = 1.00 \times 10^{-13}$ (column 5 and 8), $P = 1.53 \times 10^{-5}$ (column 6 and 8), and $P = 5.71 \times 10^{-11}$ (column 7 and 8). The different letters indicate the significant difference determined by one-way ANOVA followed by the Tukey test ($P < 0.05$). **c** Six-week-old soil-grown plants are shown. Scale bar, 1 cm (left panel) and 2 cm (right panel). **d, e** LET1 and LET2/MDS1 function additively in regulating *mkk1/2* cell death. Four-week-old soil-grown plants (**d**) and leaves (**e**) are shown. Scale bar, 1 cm. The leaves from the individual plants were placed with the order of age (from oldest to youngest). **f, g** LET1 and LET2/MDS1 function additively in regulating *mpk4* cell death. Four-week-old soil-grown plants (**f**) and their fresh weight (**g**) are shown. Scale bar, 1 cm. The data are shown as the mean ± SE ($n = 5$) with one-way ANOVA followed by the Tukey test ($P < 0.05$). $P = 7.72 \times 10^{-13}$ (column 5 and 6), $P = 3.51 \times 10^{-8}$ (column 5 and 7), $P = 4.71 \times 10^{-13}$ (column 5 and 8), $P = 4.71 \times 10^{-13}$ (column 5 and 9), and $P = 4.71 \times 10^{-13}$ (column 5 and 10). The above experiments were repeated three times with similar results.

SUMM2[ac] is comparable in WT and *let2-1*. The data indicate that LET2/MDS1 is not required for active SUMM2-triggered cell death and might act independently or upstream of SUMM2. Taken together, our results suggest that LET2/MDS1 functions genetically downstream of MEKK2 and upstream of SUMM2 in the *mekk1-mkk1/2-mpk4* cell death pathway. However, we cannot rule out the possibility that LET2/MDS1 functions independently of SUMM2 in the *mekk1-mkk1/2-mpk4* cell death pathway.

**LET2/MDS1 promotes LET1 phosphorylation and hetero-merizes with LET1.** Consistent with the genetic data, a co-immunoprecipitation (Co-IP) assay with HA-tagged LET2/MDS1, and FLAG-tagged MEKK2, SUMM2, or MPK4 co-expressing in *Arabidopsis* protoplasts indicated that LET2/MDS1 associated with MEKK2 and SUMM2, but not MPK4 (Fig. 4a). We observed an increased protein accumulation of LET2-HA when co-expressing with MEKK2-GFP, but not GFP alone, in *N. benthamiana* (Fig. 4b). Notably, MEKK2 did not affect GFP proptein level (Fig. 4b). The data suggest that MEKK2 might stabilize LET2/MDS1 in modulating SUMM2 activation. Consistently, LET2/MDS1 proteins were stabilized by the

treatment of MG132, a proteasome-dependent protein degradation inhibitor, in *35S::LET2-HA* transgenic plants and in *N. benthamiana* (Fig. 4c, d). Notably, the effect of MG132 was less pronounced in the presence of MEKK2, suggesting that MEKK2 had a similar effect with MG132 on the stabilization of LET2-HA (Fig. 4d). The defect of MEKK1-MKK1/2-MPK4 pathway induced accumulation of MEKK2 transcripts and proteins[47], which might lead to the stabilization of LET2/MDS1. Consistent with this hypothesis, the amount of LET2-HA protein was increased in three independent *35S::LET2-HA* transgenic plants upon silencing *MEKK1* by VIGS (Fig. 4e). Collectively, these results suggest that MEKK2 modulates LET2/MDS1 protein homeostasis.

Significantly, we observed a mobility shift of LET1 in the presence of LET2/MDS1, but not its kinase mutant LET2[KM] (Fig. 4f). The mobility shift of LET1 induced by LET2/MDS1 could be removed by the λ-phosphatase treatment (Fig. 4g), suggesting that LET2/MDS1 promotes LET1 phosphorylation in a kinase activity-dependent manner. Apparently, LET2/MDS1 did not induce mobility shift of FER (Supplementary Fig. 3a), and FER also did not affect LET1 mobility (Supplementary Fig. 3b). We also did not observe any mobility shift of LET2/MDS1 in the

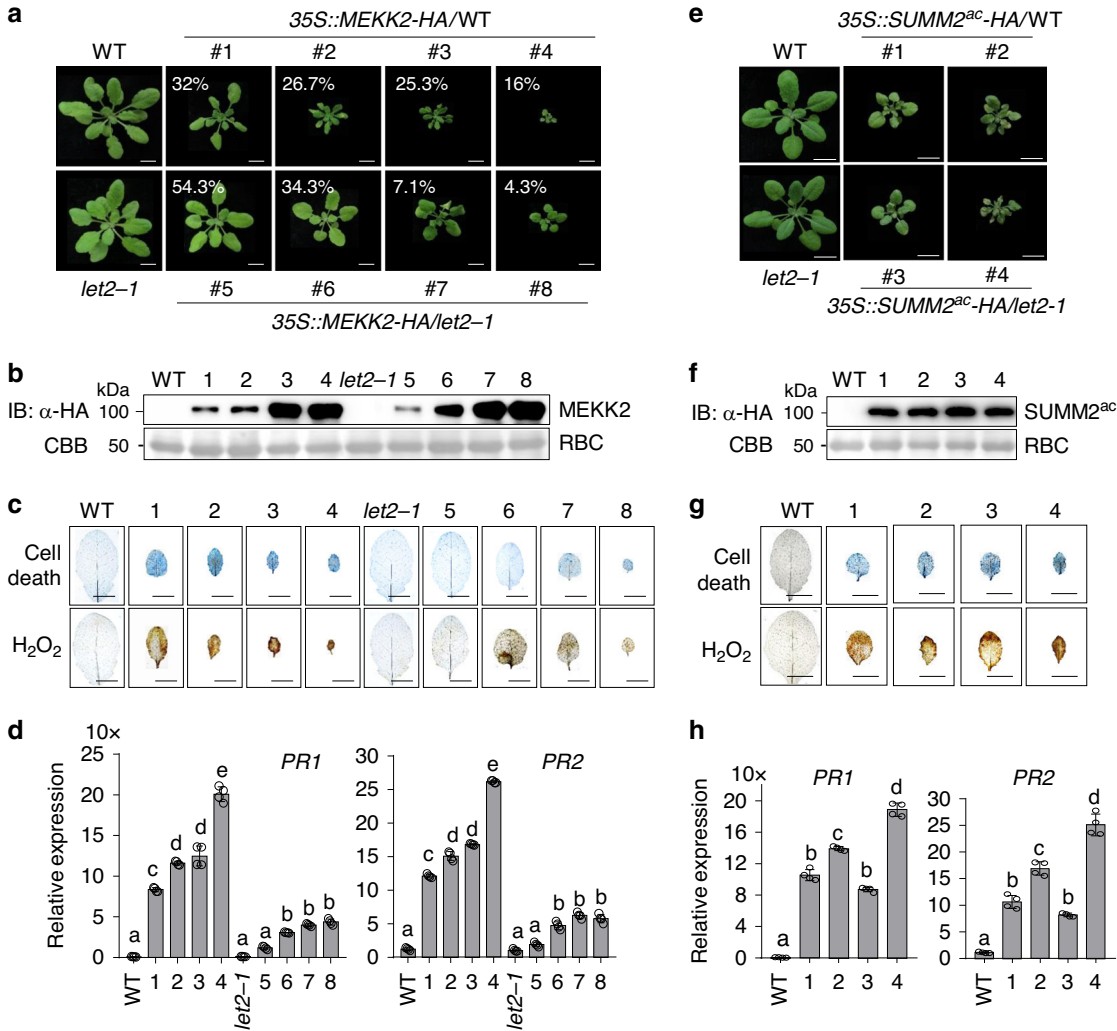

**Fig. 3 LET2/MDS1 is required for MEKK2-, but not SUMM2$^{ac}$-mediated autoimmunity. a** Plant dwarfisms and growth defects triggered by overexpressing *MEKK2* in WT are alleviated in the *let2-1* mutant. 75 and 70 independent primary (T$_1$) transgenic plants carrying *35S::MEKK2-HA* in WT and *let2-1* were characterized, respectively. Four-week-old plants representing different levels of dwarfisms labeled with the percentage of the cognate category are shown. Scale bar, 1 cm. **b** Protein expression of MEKK2-HA in transgenic plants. Total proteins were isolated from plants in **a** and immunoblotted using an α-HA antibody (top panel). CBB staining for RBC is shown as the loading control (bottom panel). **c** The cell death and H$_2$O$_2$ accumulation triggered by overexpressing *MEKK2* in WT are reduced in the *let2-1* mutant. Leaves from plants in **a** were stained by trypan blue for cell death (upper panel) and DAB for H$_2$O$_2$ (lower panel). Scale bar, 0.5 cm. **d** The elevated expression of *PR1* and *PR2* triggered by overexpressing *MEKK2* in WT is reduced in *let2-1*. The expression of *PR1* and *PR2* was normalized to the expression of *UBQ10* and the data are shown as the mean ± SE of four biological repeats (*n* = 4). The different letters indicate the significant difference determined by one-way ANOVA followed by the Tukey test (*P* < 0.05). The plants 1–4 are *35S::MEKK2-HA*/WT, and 5–8 are *35S::MEKK2-HA/let2-1* (**a–d**). **e** Plant dwarfisms and growth defects triggered by overexpressing *SUMM2$^{ac}$* are similar in WT and *let2-1*. In all, 68 and 71 independent primary (T$_1$) transgenic plants carrying *35S::SUMM2$^{ac}$-HA* in WT and *let2-1* were characterized respectively. Two representastive 3-week-old plants, which showed the growth defects, and their controls, are shown in the figure. Scale bar, 1 cm. **f** Protein expression of SUMM2$^{ac}$-HA in transgenic plants. **g** The cell death and H$_2$O$_2$ accumulation triggered by overexpressing *SUMM2$^{ac}$* in WT and *let2-1*. **h** The expression levels of *PR1* and *PR2* triggered by overexpressing *SUMM2$^{ac}$* in WT and *let2-1*. The expression of *PR1* and *PR2* was normalized to the expression of *UBQ10* and the data are shown as the mean ± SE of four biological repeats (*n* = 4). *P* < 1.00 × 10$^{-15}$ (*PR1*, column 1 and 2), *P* < 1.00 × 10$^{-15}$ (*PR1*, column 1 and 3), *P* = 2.62 × 10$^{-12}$ (*PR1*, column 1 and 4), *P* < 1.00 × 10$^{-15}$ (*PR1*, column 1 and 5), *P* = 1.24 × 10$^{-7}$ (*PR2*, column 1 and 2), *P* = 1.04 × 10$^{-10}$ (*PR2*, column 1 and 3), *P* = 5.86 × 10$^{-6}$ (*PR2*, column 1 and 4), *P* < 1.00 × 10$^{-15}$ (*PR2*, column 1 and 5). The different letters indicate the significant difference determined by one-way ANOVA followed by the Tukey test (*P* < 0.05). The plants 1–2 are *35S::SUMM2$^{ac}$-HA*/WT, and 3–4 are *35S::SUMM2$^{ac}$-HA/let2-1* (**e–h**). The above experiments were repeated three times with similar results.

presence of LET1 in either regular or Phos-tag SDS-PAGE (Supplementary Fig. 3c). The data indicate that LET2/MDS1 specifically promotes LET1 phosphorylation. Consistently, LET2/MDS1, not its kinase-inactive mutant, activated the kinase activity of LET1 in vitro when LET1 and LET2/MDS1 were co-expressed in protoplasts and immunoprecipitated for an in vitro kinase assay (Fig. 4h). We further observed that LET2/MDS1 complexed with LET1 in a Co-IP assay (Fig. 4i). The Förster resonance

energy transfer (FRET)-fluorescence lifetime imaging (FLIM) measurements revealed that LET1-GFP proteins were in close proximity to LET2-mCherry, but not BIR2-mCherry, when co-expressed in *Arabidopsis* protoplasts (Fig. 4j, k). Furthermore, LET2/MDS1 extracellular domain (LET2$^{ex}$) purified from *E. coli* could pull-down LET1-FLAG expressed in protoplasts (Fig. 4l). Thus, LET2/MDS1 complexes with LET1 and regulates LET1 phosphorylation.

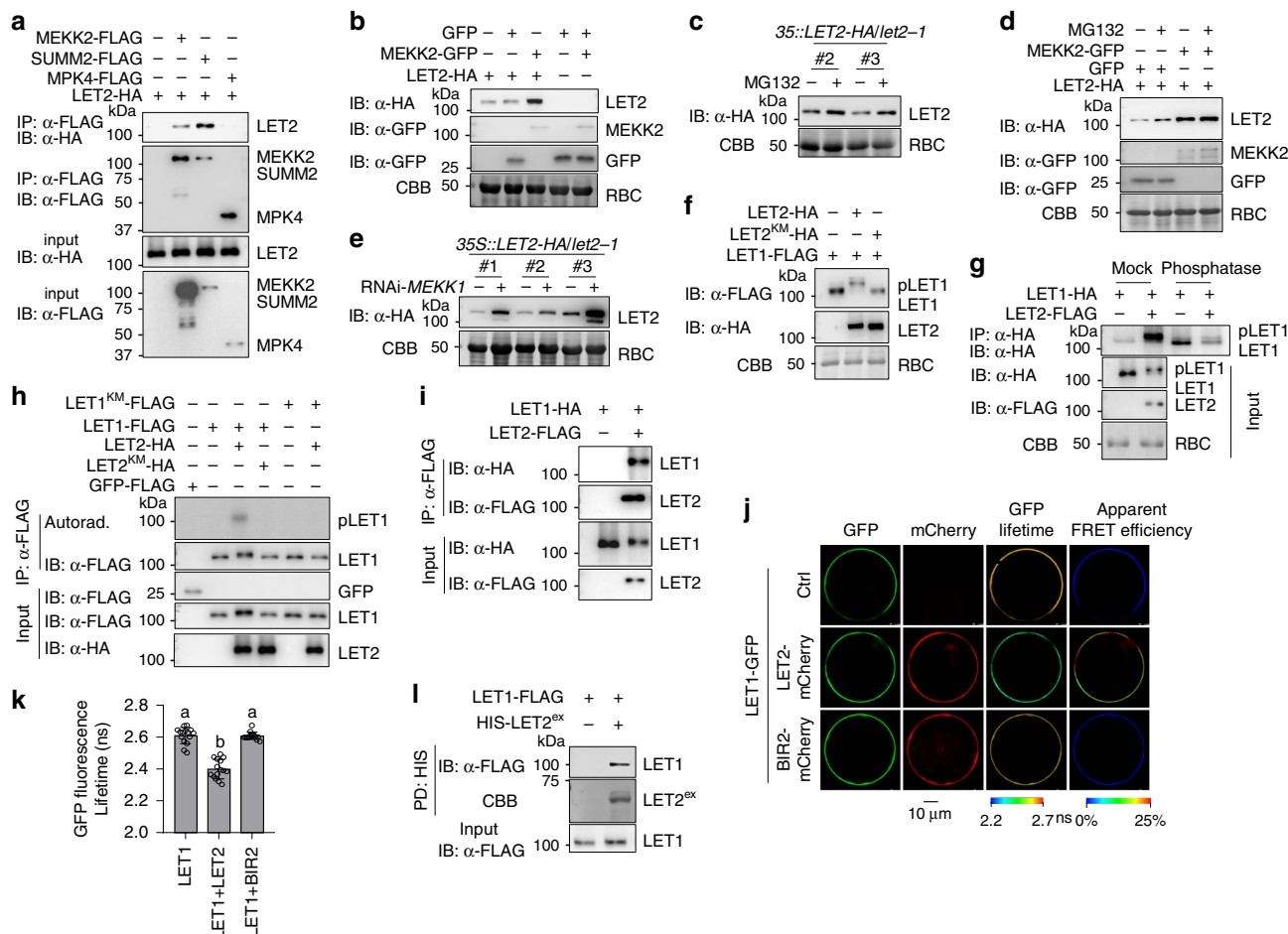

**Fig. 4 LET1 and LET2/MDS1 regulation and heteromerization. a** LET2/MDS1 associates with MEKK2 and SUMM2, but not MPK4. LET2-HA was co-expressed with Ctrl, MEKK2-FLAG, SUMM2-FLAG, or MPK4-FLAG in protoplasts for 12 h. The FLAG-tagged proteins were immunoprecipitated by α-FLAG affinity beads, and then immunoblotted by an α-HA or α-FLAG antibody (top two panels). The proteins before immunoprecipitation were immunoblotted by an α-HA or α-FLAG antibody as inputs (bottom two panels). **b** MEKK2 stabilizes LET2/MDS1 protein accumulation in *N. benthamiana*. LET2-HA or GFP was co-expressed with MEKK2-GFP or GFP in *N. benthamiana* for 3 days. The proteins were immunoblotted by an α-HA or α-GFP antibody. CBB staining of RBC was used as a loading control. **c** MG132 treatment increases LET2/MDS1 protein accumulation in transgenic plants. The 10-day-old seedings of *35S::LET2-HA/ let2-1* (Line #2 and #3) transgenic plants were treated with DMSO (Ctrl) or 5 μM MG132 for 6 h. Proteins were immunoblotted using an α-HA antibody, and CBB was used as a loading control. **d** MG132 treatment increases LET2/MDS1 protein accumulation in *N. benthamiana*. The leaves of *N. benthamiana* were inoculated with Agrobacterium carrying *LET1-HA* and *GFP*, or *LET1-HA* and *MEKK2-GFP* for 12 h, and then treated with DMSO (Ctrl) or 5 μM MG132 for anthor 36 h. Proteins were immunoblotted by an α-HA or α-GFP antibody. CBB was used as a loading control. **e** Silencing *MEKK1* increases LET2/MDS1 protein accumulation. *MEKK1* was silenced in the *35S::LET2-HA/let2-1* transgenic plants (Line #1, #2 and #3) by VIGS. Total proteins were extracted 2 weeks after VIGS, and immunoblotted using an α-HA antibody. CBB was used as a loading control. **f** LET2/MDS1, but not its kinase-inactive mutant LET2$^{KM}$, induces LET1 mobility shift. LET2-HA or LET2$^{KM}$-HA was co-expressed with LET1-FLAG in protoplasts for 12 h. LET1-FLAG was separated by 7.5% SDS-PAGE. CBB staining of RBC was used as a loading control. **g** LET2/MDS1 induces LET1 phosphorylation. LET1-HA was co-expressed with Ctrl or LET2-FLAG in protoplasts for 12 h. LET1-HA was immunoprecipitated by α-HA affinity beads. The immunoprecipitated LET1-HA protein was incubated without or with 0.5 μL (200 U) λ-phosphatase (Sigma) for 1 h at 30 °C. LET1-HA was separated by 10% SDS-PAGE and detected by an α-HA antibody (top panel). LET1-HA and LET2-FLAG before immunoprecipitation were detected by the corresponding antibody (middle two panels). CBB staining of RBC was used as a loading control (bottom panel). **h** LET2/MDS1 increases LET1 kinase activity. LET1-FLAG or LET1$^{KM}$-FLAG was co-expressed with the vector control, LET2-HA or LET2$^{KM}$-HA, in protoplasts. The FLAG-tagged proteins were immunoprecipitated from the cell lysates with α-FLAG affinity beads and used in a kinase assay with [γ-$^{32}$P] ATP. The GFP-FLAG was used as a negative control. The proteins were immunoblotted by an α-FLAG or α-HA antibody for input controls. **i** LET1 associates with LET2/MDS1. LET1-HA was co-expressed with Ctrl or LET2-FLAG in protoplasts for 12 h. The LET2-FLAG proteins were immunoprecipitated by α-FLAG affinity beads, and then immunoblotted by an α-HA or α-FLAG antibody (top two panels). The proteins before immunoprecipitation were immunoblotted by an α-HA or α-FLAG antibody as inputs (bottom two panels). **j, k** FRET-FLIM analysis of LET1 and LET2/MDS1 interaction in *Arabidopsis* protoplasts. The indicated proteins were transiently expressed in protoplasts for 16 h, and FRET-FLIM was visualized using a confocal laser scanning microscopy (**j**). Localization of the LET1-GFP and LET2-mCherry/BIR2-mCherry is shown with the first (Green) and second column (Red), respectively. The lifetime (τ) distribution (third column), and apparent FRET efficiency (fourth column) are presented as pseudo-color images according to the scale. The GFP mean fluorescence lifetime (τ) values, ranging from 2.2 to 2.7 nanoseconds (ns), were statistically analyzed and are shown as mean ± SD ($n = 15$) (**k**). $P = 1.07 \times 10^{-12}$ (column 1 and 2), $P = 1.08 \times 10^{-12}$ (column 2 and 3). The different letters indicate the significant difference determined by one-way ANOVA followed by the Tukey test ($P < 0.05$). Scale bar, 10 μm. **l** LET2$^{ex}$ associates with LET1 in a pull-down assay. *Arabidopsis* protoplasts expressing LET1-FLAG were incubated with purified HIS-SUMO-LET2$^{ex}$ proteins. The interaction between LET1 and LET2$^{ex}$ was detected by an α-FLAG immunoblot after pull-down with Ni-NTA agarose. HIS-SUMO-LET2$^{ex}$ proteins were stained by CBB. The above experiments were repeated three times with similar results.

**The *llg1* mutants specifically suppress RNAi-*MEKK1* cell death.** The GPI-anchored proteins LRE, LLG1, LLG2, and LLG3, function as adapters/co-receptors for *Cr*RLK1Ls FER and BUPSs/ANXs[28–31]. LLG2 and LLG3 function redundantly in regulating pollen tube integrity[30,31]. We tested whether LRE/LLGs are involved in LET1/2-mediated *mekk1* cell death by silencing *MEKK1* in the corresponding single and double mutants, including two *lre* mutant alleles (*lre-3* and *lre-6*), two *llg1* mutant alleles (*llg1-1* and *llg1-2*), *llg2-1*, and *llg3-1* single mutants, and *llg2-1llg3-1* double mutant. The *llg1* mutants, *llg1-1* and *llg1-2*, but not other mutants, suppressed the growth defects (Fig. 5a and Supplementary Fig. 4a), cell death, $H_2O_2$ accumulation (Fig. 5b), and constitutive expression of *PR* genes (Supplementary Fig. 4b) caused by silencing *MEKK1*, suggesting a specific role of LLG1 in controlling *mekk1* cell death. Both *llg1-1* and *llg1-2* mutants display certain growth defects with reduced plant size (Fig. 5a). The *llg1-3* mutant, which bears a mutation in glycine at the 114th amino acid to arginine, grew similarly as WT plants (Supplementary Fig. 4c)[32]. To exclude the effect of growth defect of *llg1-1* and *llg1-2* on RNAi-*MEKK1* cell death, we silenced *MEKK1* in the *llg1-3* mutant by VIGS. The *llg1-3* mutant also suppressed growth retardation, cell death and *PR1* expression caused by RNAi-*MEKK1* (Supplementary Fig. 4c, d), indicating that LLG1-regulated growth and MEKK1 cell death are uncoupled. We also tested whether LLG1 with a N-terminal HA tag under its native promoter in *llg1-2* (*pLLG1::HA-LLG1/llg1-2*)[28] could complement RNAi-*MEKK1* cell death. Two representative lines, #1 and #2, restored the cell death and *PR* gene expression induced by silencing *MEKK1* (Fig. 5c, d and Supplementary Fig. 4b). The data implicate that similar to LET1 and LET2/MDS1, LLG1 modulates RNAi-*MEKK1* cell death.

We then generated the *llg1-1mekk1* double mutant by genetic crosses. To our surprise, the *llg1-1mekk1* double mutant did not suppress *mekk1* cell death. In contrast, the *llg1-1mekk1* mutant displayed more severe growth defects, reduced fresh weight and elevated *PR1* gene expression than the *mekk1* mutant of 3-week-old seedlings grown on ½MS plates (Fig. 5e–g). To rule out the allele specific effect of *llg1-1*, we generated *llg1-2mekk1* and *llg1-3mekk1* double mutants. Similar to *llg1-1mekk1*, both *llg1-2mekk1* and *llg1-3mekk1* mutants displayed further aggravated growth defects compared to *mekk1* (Supplementary Fig. 5a, b). The data suggest that the *llg1* mutants can enhance the growth defects of genetic null mutant *mekk1*. The *mekk1-mkk1/2-mpk4* cell death is temperature dependent, and the moderately increased temperature could alleviate *mekk1* cell death likely through the suppression of the salicylic acid (SA) pathway[44]. However, unlike *mekk1* or *llg1-1*[+/−]*mekk1* (*llg1-1* is heterozygous), the growth defects of *llg1-1mekk1* grown at 28 °C were as severe as plants grown at 22 °C (Supplementary Fig. 5c). The data suggest that the enhanced growth defects of *llg1mekk1* is temperature-independent, which is different from that of *mekk1*.

The above data point to an intriguing and apparently contradict observation: the *llg1* mutants aggravated the growth defects caused by genetic lesions in *mekk1*, whereas suppressed cell death triggered by RNAi-mediated silencing of *MEKK1*. Notably, VIGS was performed using 12-day-old seedlings, and the silence effects of *MEKK1* were apparent after 20-days post-germination. Compared to the genetic mutations, VIGS-mediated silencing bypasses the defects associated with embryonic and early seedling development[51]. The opposing effects of *LLG1* on silencing and null mutations of *MEKK1* suggest that *LLG1* plays one role in regulating initial seedling development in concert with MEKK1, and another role in regulating *mekk1* cell death at a later stage. This is in line with the notion that LLG1 acts through interactions with different *Cr*RLK1Ls as an adapter/co-receptor and regulates various biological processes. The mutations of

LLG1 and the *Cr*RLK1L FER cause similar growth defects[28]. We tested whether the *fer-4* mutant exerted an effect on the *mekk1* growth defects by generating a *fer-4mekk1* double mutant. Similar to *llg1mekk1*, the *fer-4mekk1* double mutant showed further aggravated growth defects (Fig. 5h), reduced fresh weight (Fig. 5i), and increased *PR1* gene expression (Fig. 5j) compared to the *mekk1* mutant. Unlike *mekk1*, the enhanced growth defects in the *fer-4mekk1* mutant cannot be recovered when plants were grown at 28 °C (Fig. 5k, l), consistent with the temperature-independent cell death in the *llg1-1mekk1* mutants (Supplementary Fig. 5c). Taken together, LLG1 is required for *mekk1* cell death. In addition, LLG1 plays a role together with FER and MEKK1 in regulating early seedling development.

**The *llg1-1* mutation suppresses *mkk1/2* and *mpk4* cell death.** We further tested whether the mutation in *LLG1* affects *mkk1/2* and *mpk4* cell death by generating the *llg1-1mkk1/2* (*llg1mkk1/2*) triple mutant (Supplementary Fig. 6a), and the *llg1-1mpk4* (*llg1mpk4*) double mutant (Supplementary Fig. 6b). The *llg1-1* mutant partially suppressed the cell death in the *mkk1/2* mutant when grown on ½MS plates (Fig. 6a). The *llg1mkk1/2* mutant was bigger than *mkk1/2* in size and had significantly increased fresh weight compared to *mkk1/2* (Fig. 6b). The true leaves of *llg1mkk1/2* were also larger than those of *mkk1/2*. At the 2-week-old seedling stage, the first pair of true leaves already senescenced in *mkk1/2* but still kept green in *llg1mkk1/2* (Fig. 6a). Compared with *mkk1/2*, the expression of *PR1* was partially suppressed in *llg1mkk1/2* (Fig. 6c). Notably, the *llg1*[+/−] *mkk1/2* mutant, in which *LLG1* was heterozygous, had no effect on *mkk1/2* cell death, suggesting that *llg1* is a complete recessive mutation in regulating *mkk1/2* cell death.

The rosette size of the *llg1mpk4* mutant was also bigger than *mpk4* when grown on soil (Fig. 6d). The fresh weight of *llg1mpk4* was significantly higher than that of *mpk4* (Fig. 6e), and the increased *PR1* expression in *mpk4* was partially reduced in *llg1mpk4* (Fig. 6f). Interestingly, the *llg1*[+/−]*mpk4* mutant behaved in between *mpk4* and *llg1mpk4* in terms of rosette size, fresh weight, and *PR1* gene expression (Fig. 6d–f), indicating that *LLG1* regulates *mpk4* cell death in a dosage-dependent manner. It was reported that the *mekk2* mutant rescued the *mpk4* cell death in a dosage-dependent manner[47]. Altogether, similar as the *let2* mutants, the *llg1* mutants suppressed *mkk1/2* and *mpk4* cell death, suggesting that LLG1 functions genetically downstream of MPK4 in regulating *mekk1-mkk1/2-mpk4* cell death.

**The mutations in *LLG1* block MEKK2-, but not SUMM2[ac]-triggered cell death.** To delineate the genetic position of LLG1 with MEKK2 and SUMM2 in the regulation of cell death, we examined whether *llg1* mutants exerted an effect on over-expressing *MEKK2* or active SUMM2 (SUMM2[ac])-triggered cell death by expressing *35S::MEKK2-HA* or *35S::SUMM2[ac]-HA* in *llg1* mutants. As shown previously (Fig. 3a), overexpressing *MEKK2-HA* in WT caused growth defects and elevated *PR1* expression in a dosage-dependent manner (Fig. 6g–i). However, multiple transgenic lines expressing *35S::MEKK2-HA* in *llg1-1* were phenotypically similar to *llg1-1* in terms of plant size irregardless of MEKK2-HA protein expression levels (Fig. 6g, h). The increased expression of *PR1* triggered by *35S::MEKK2-HA* in WT plants was also reduced in *llg1-1* (Fig. 6i). In addition, another *LLG1* mutant allele, *llg1-3*, also blocked overexpressing *MEKK2-HA*-triggered growth defects and cell death (Supplementary Fig. 7a, b). The data indicate that LLG1 is required for MEKK2-triggered cell death and acts genetically downstream of MEKK2. However, the *llg1-3* mutant did not affect growth defects and cell death caused by overexpressing *SUMM2[ac]-HA* compared to WT

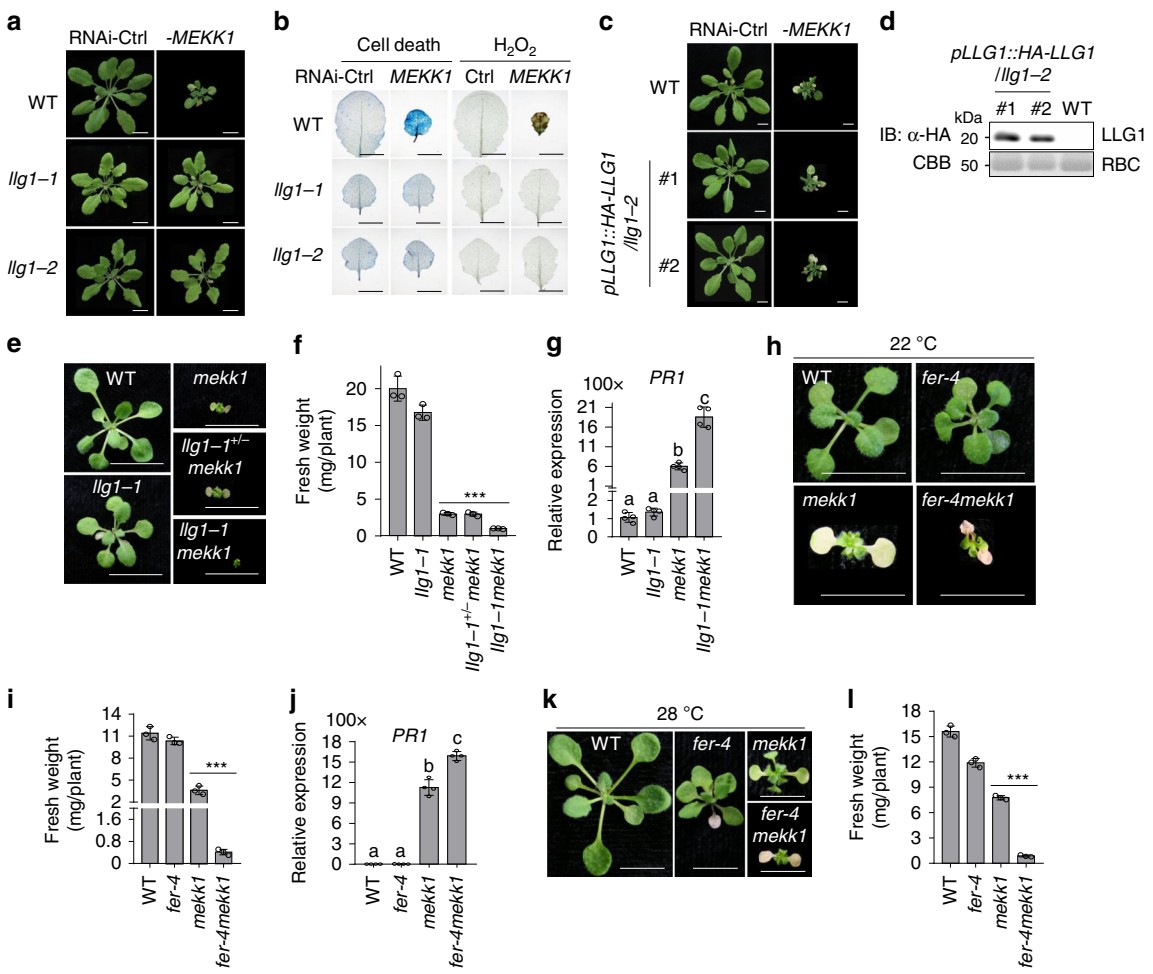

**Fig. 5 LLG1 regulates *mekk1* cell death. a** The *llg1* mutants suppress growth defects triggered by silencing *MEKK1*. The plant images were photographed at 3-weeks after inoculation with Agrobacterium carrying the indicated VIGS vectors. Ctrl is the vector containing GFP. Scale bar, 1 cm. **b** The *llg1* mutants suppress cell death and H$_2$O$_2$ accumulation induced by silencing *MEKK1*. The leaves from plants in **a** were stained by trypan blue for cell death and DAB for H$_2$O$_2$ accumulation. Scale bar, 0.5 cm. **c** Expression of *HA-LLG1* in *llg1-2* restores the cell death triggered by silencing *MEKK1*. #1 and #2 are two representative *pLLG1::HA-LLG1* transgenic lines in *llg1-2*. Scale bar, 0.5 cm. **d** Protein expression of HA-LLG1 in *pLLG1::HA-LLG1/llg1-2* transgenic plants. **e** The *llg1-1mekk1* mutant enhances growth defects of *mekk1*. The seedlings grown on ½MS plate at 22 °C were photographed at 2-weeks post-germination. Scale bar, 0.5 cm. **f** The fresh weight of *llg1-1mekk1* is less than *mekk1*. The data are shown as mean ± SE ($n = 3$). $P = 3.63 \times 10^{-5}$ (column 3 and 5). The asterisk indicates statistical significance by using two-sided two-tailed Student's *t* test (***$P < 0.001$). **g** *llg1-1* mutant enhances the expression of *PR1* in *mekk1*. The expression of *PR1* was determined with the plants in **e** and normalized to the expression of *UBQ10*. The data are shown as the mean ± SE of four biological repeats ($n = 4$). $P = 1.04 \times 10^{-7}$ (column 3 and 4). The different letters indicate the significant difference determined by one-way ANOVA followed by the Tukey test ($P < 0.05$). **h** The *fer-4mekk1* mutant enhances growth defects of *mekk1*. The seedlings grown on ½MS plate at 22 °C were photographed at 2-weeks post-germination. Scale bar, 1 cm. **i** The fresh weight of *fer-4mekk1* mutant is less than *mekk1*. The data are shown as mean ± SE ($n = 3$). $P = 6.92 \times 10^{-4}$ (column 3 and 4). The asterisk indicates statistical significance by using two-sided two-tailed Student's *t* test (***$P < 0.001$). **j** *fer-4* mutant enhances the expression of *PR1* in *mekk1*. The expression of *PR1* was normalized to the expression of *UBQ10* and the data are shown as the mean ± SE of four biological repeats ($n = 4$). $P = 1.87 \times 10^{-6}$ (column 3 and 4). The different letters indicate the significant difference determined by one-way ANOVA followed by the Tukey test ($P < 0.05$). The assay was performed as in **g**. **k** High temperature did not alleviate *fer-4mekk1* growth defects. The seedlings grown on ½MS plate at 28 °C were photographed at 2-weeks post-germination. Scale bar, 1 cm. **l** The fresh weight of *fer-4mekk1* mutant is less than *mekk1* at 28 °C. The seedlings in **k** were used for measuring fresh weight. The data are shown as mean ± SE ($n = 3$). $P = 1.81 \times 10^{-6}$ (column 3 and 4). The asterisk indicates statistical significance by using two-sided two-tailed Student's *t* test (***$P < 0.001$). To measure the weight of *mekk1*, *llg1mekk1*, and *fer-4mekk1* mutants, 10 plants were pooled and the weight of individual plants was averaged. The above experiments were repeated 3–4 times with similar results.

plants (Supplementary Fig. 7c, d). Taken together, similar to LET1 and LET2/MDS1, LLG1 functions downstream of MEKK2 and upstream of SUMM2 in the *mekk1-mkk1/2-mpk4* cell death pathway.

**LLG1 associates with LET proteins and is required for their plasma membrane localization.** LLGs directly interacts with the extracellular juxtamembrane region of some *Cr*RLK1Ls and function as co-receptors/adapters of *Cr*RLK1Ls in regulating plant growth, reproduction, and immunity[28–31]. We hypothesized that LLG1 functions in the *mekk1-mkk1/2-mpk4* cell death pathway through interaction and modulation of *Cr*RLK1Ls LET1 and LET2/MDS1. To test this, we performed Co-IP assays between LLG1 and LET1 or LET2/MDS1. When N-terminal HA-tagged LLG1 (HA-LLG1) was co-expressed with C-terminal

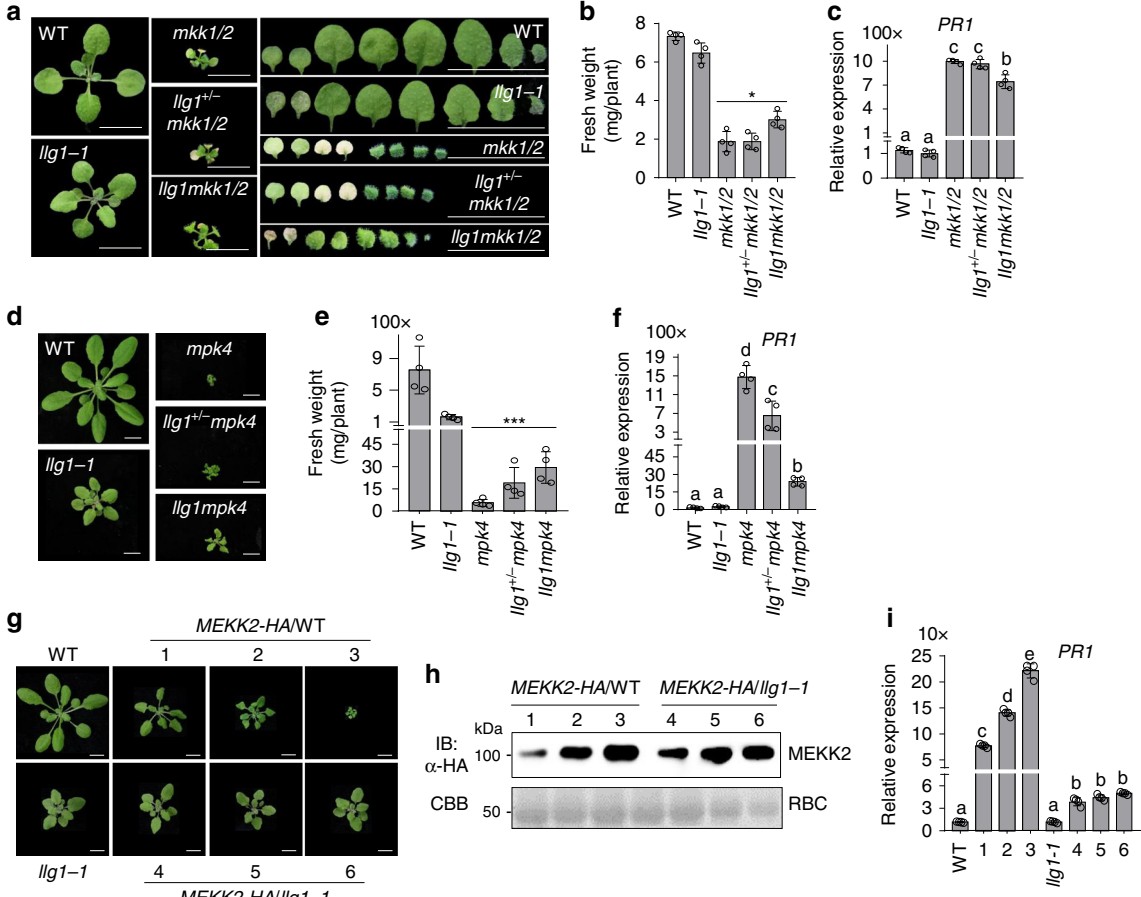

**Fig. 6 LLG1 genetically functions downstream of MEKK2. a** The *llg1-1* mutant partially suppresses the *mkk1/2* cell death. The seedlings grown on ½MS plate were photographed at 2-weeks post-germination. The leaves from the individual plants were placed with the order of age (from oldest to youngest, right panels). Scale bar, 0.5 cm. **b** *llg1-1* increases the fresh weight of *mkk1/2*. The data are shown as mean ± SE ($n = 4$). $P = 1.46 \times 10^{-2}$ (column 3 and 5). The asterisk indicates statistical significance by using two-sided two-tailed Student's $t$ test (*$P < 0.05$). **c** The *llg1-1* mutant reduces the expression of *PR1* in *mkk1/2*. The expression of *PR1* was determined with the plants in **a** and normalized to the expression of *UBQ10*. The data are shown as the mean ± SE of four biological repeats ($n = 4$). $P = 2.06 \times 10^{-5}$ (column 3 and 5). The different letters indicate the significant difference determined by one-way ANOVA followed by the Tukey test ($P < 0.05$). **d** The *llg1-1* mutant partially suppresses the *mpk4* cell death. The plants grown on soil were photographed at 4-weeks after germination. Scale bar, 1 cm. **e** *llg1-1* increases the fresh weight of *mpk4*. The data are shown as mean ± SE ($n = 4$). $P = 4.69 \times 10^{-3}$ (column 3 and 5). The asterisk indicates statistical significance by using two-sided two-tailed Student's $t$ test (***$P < 0.01$). **f** The *llg1-1* mutant reduces the expression of *PR1* in *mpk4*. The expression of *PR1* was determined with the plants in **d** and normalized to the expression of *UBQ10*. The data are shown as the mean ± SE of four biological repeats ($n = 4$). $P = 6.78 \times 10^{-8}$ (column 3 and 5). The different letters indicate the significant difference determined by one-way ANOVA followed by the Tukey test ($P < 0.05$). **g** Overexpressing *MEKK2* triggers growth defects in WT, but not in *llg1-1*. Three representative lines were used to indicate the phenotype of transgenic lines in WT Col-0 and *llg1-1* with different MEKK2-HA expression level. The pictures were taken with 4-week-old soil-grown plants. Scale bar, 1 cm. **h** The protein accumulation of MEKK2-HA in WT and *llg1-1* transgenic plants. Total proteins were isolated from plants in **g** and immunoblotted using an α-HA antibody (top panel). CBB staining of RBC is shown as the loading control (bottom panel). **i** The elevated expression of *PR1* triggered by overexpressing *MEKK2* in WT (Lines 1, 2, and 3) is reduced in *llg1-1* (Lines 4, 5, and 6). The expression of *PR1* was normalized to the expression of *UBQ10* and the data are shown as the mean ± SE of four biological repeats ($n = 4$). The different letters indicate the significant difference determined by one-way ANOVA followed by the Tukey test ($P < 0.05$). To measure the weight of *mkk1/2* and *mpk4* mutants, 10 plants were pooled and the weight of individual plants was averaged. The above experiments were repeated three times with similar results.

FLAG-tagged LET1 (LET1-FLAG) or LET2/MDS1 (LET2-FLAG) in *Arabidopsis* protoplasts, both LET1 and LET2/MDS1 immunoprecipitated LLG1 (Fig. 7a). We further tested whether LLG1 interacted with the extracellular malectin-like domains (ECD) of LET1 (LET1$^{ECD}$). When co-expressed in *Arabidopsis* protoplasts, LET1$^{ECD}$-FLAG could co-immunoprecipitate HA-LLG1 (Fig. 7b). We then determined whether LLG1 directly interacted with the extracellular juxtamembrane (exJM) domain of LET1 with an in vitro pull-down assay. To do this, we purified the exJM domain of LET1 (amino acid 337–400) fused with glutathione S-transferase (GST-LET1$^{exJM}$), and the LLG1 truncation without the signal peptide (SP; amino acid 24–149) fused with the

maltose-binding protein (MBP-LLG1) from *E. coli* and performed an in vitro pull-down assay with glutathione agarose beads. As shown in Fig. 7c, GST-LET1$^{exJM}$, but not GST alone, pulled down MBP-LLG1, indicating a direct interaction between LLG1 and the exJM domain of LET1. Taken together, LLG1 likely functions as an adapter/co-receptor of LET1 and LET2/MDS1 in cell death regulation.

LLG1 functions as a chaperone assisting FER protein delivery from the endoplasmic reticulum (ER) to the plasma membrane (PM), which is essential for extracellular signal perception and signaling initiation[28]. To test where LET1 and LET2/MDS1 are localized and whether the localization is mediated by LLG1, we

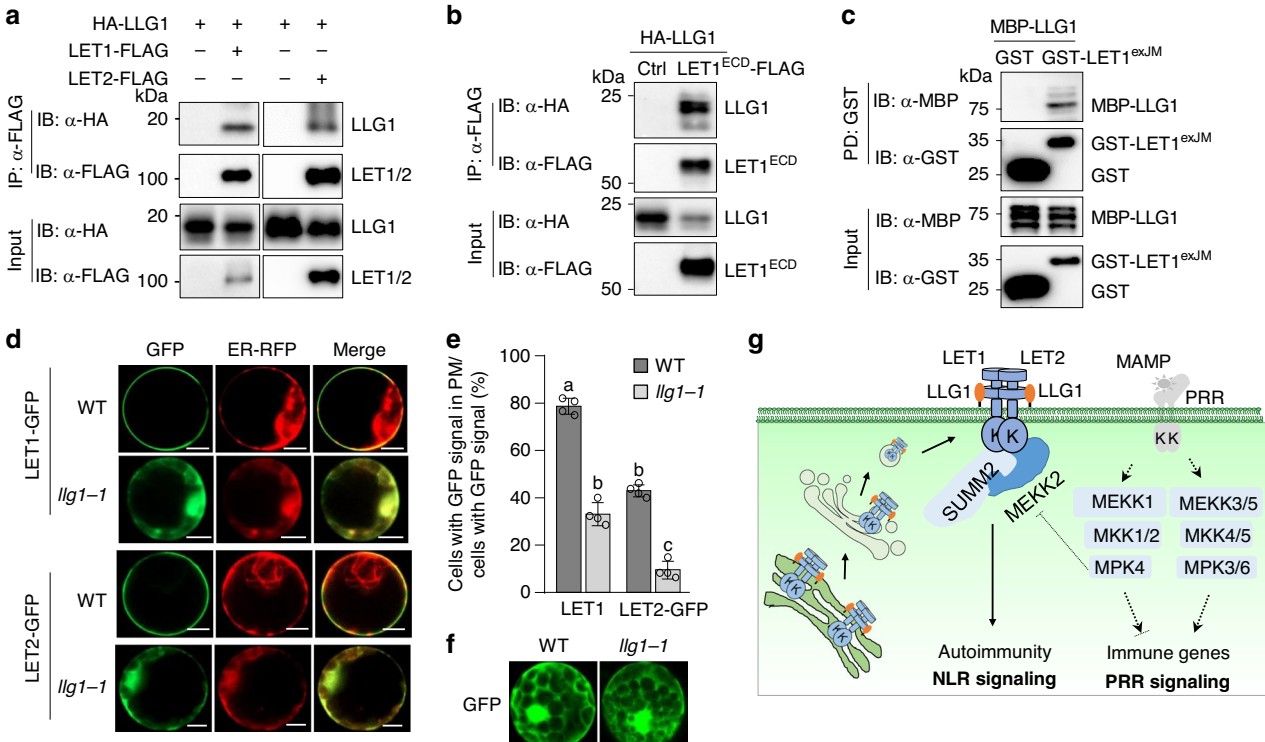

**Fig. 7 LLG1 interacts with LETs and facilitates their plasma membrane localization. a** LLG1 interacts with LET1 and LET2/MDS1. HA-LLG1 was co-expressed with Ctrl, LET1-FLAG or LET2-FLAG, in *Arabidopsis* protoplasts. Total proteins were immunoprecipitated with α-FLAG affinity beads and then immunoblotted by an α-HA or α-FLAG antibody (top two panels). Immunoblots using total proteins before immunoprecipitation are shown as protein inputs (bottom two panels). **b** LLG1 interacts with the LET1 extracellular domain (LET1$^{ECD}$) in *Arabidopsis* protoplasts. The assay was performed as in **a**. **c** LLG1 directly interacts with LET1 extracellular juxtamembrane domain (LET1$^{exJM}$) in vitro. MBP-LLG1 (without signal peptide), GST-LET1$^{exJM}$ and GST fusion proteins isolated from *E. coli* were used for an in vitro pull-down assay using glutathione agarose beads followed by immunoblotting using an α-MBP or α-GST antibody (top two panels). Input proteins were immunoblotted by an α-MBP or α-GST antibody before pull-down (bottom two panels). **d** Representative subcellular localization patterns of LET1-GFP and LET2-GFP proteins in WT and *llg1-1* protoplasts. The images were taken under laser scanning confocal microscopy at 12 h after transfection. ER-RFP was used as an ER marker. Scale bar, 50 μm. **e** The percentage of LET1-GFP and LET2-GFP signals in PM in WT and *llg1-1*. The ratio of protoplasts with only PM-localized LET-GFP signals to the total protoplasts with GFP signals was measured over 100 fluorescent cells. The protoplasts without clear PM-localized GFP signals show ER-localized GFP signals. The data are shown as mean ± SE of four independent transfections (*n* = 4). *P* = 4.40 × 10$^{-9}$ (column 1 and 2) and *P* = 1.39 × 10$^{-7}$ (column 3 and 4). The different letters indicate the significant difference determined by one-way ANOVA followed by the Tukey test (*P* < 0.001). **f** Similar localization of GFP in WT and *llg1-1* protoplasts. The images were taken at 12 h after transfection. Scale bar, 100 μm. **g** A model of LET1-LET2-LLG1 complex in cell death regulation. MAMP-activated MEKK1-MKK1/2-MPK4 cascade regulates PRR-mediated immune signaling, and SUMM2-mediated autoimmunity via suppressing MEKK2 expression. Two *Cr*RLK1Ls, LET1 and LET2/MDS1, together with GPI-anchored protein LLG1, form a trimeric complex to modulate SUMM2 activation. LLG1 likely functions as a co-receptor of LET1/2 and assists LET1/2 protein maturation and delivery from endoplasmic reticulum (ER), Golgi to plasma membrane (PM). The above experiments were repeated three times with similar results.

analyzed the subcellular localization of LET1 fused with green fluorescent protein (LET1-GFP) and LET2-GFP in protoplasts of WT and the *llg1-1* mutant. In WT protoplasts, the majority of cells with LET1-GFP signals (~80%) displayed PM localization (Fig. 7d, e), whereas in *llg1-1*, only ~35% of cells with LET1-GFP signals displayed PM localization. The majority of LET1-GFP signals in *llg1-1* co-localized with the ER marker (Fig. 7d). Similarly, the ratio of LET2-GFP signals in PM was reduced from ~45% in WT to ~8% in *llg1-1*, and the ER localization of LET2-GFP was significantly increased in *llg1-1* (Fig. 7d, e). The *llg1-1* mutant did not affect free GFP localization (Fig. 7f). These data indicate that LLG1 is important for LET1 and LET2/MDS1 transport from ER to PM (Fig. 7g).

## Discussion

*Cr*RLK1Ls that carry an extracellular malectin-like domain are key regulators in various developmental processes and plant defense responses to pathogens[12–15]. In a parallel study of using a VIGS-based RNAi screen of *mekk1* cell death suppressors with a

collection of *Arabidopsis* T-DNA insertion lines, we identified the uncharacterized *Cr*RLK1L, LET1, as a specific regulator of *mekk1-mkk1/2-mpk4* autoimmunity[49]. In this study, by screening individual *Cr*RLK1Ls and revealed that LET2/MDS1 plays an additive role with LET1 in regulating *mekk1-mkk1/2-mpk4* autoimmunity. The LET2/MDS1 closest tandemly arrayed homologs, MDS2, MDS3, and MDS4, had little contribution in modulating this process despite of their partially redundant role in regulating plant responses to ion metal (Fig. 1d).

Similar with LET1, LET2/MDS1 acts genetically downstream of MEKK2 and upstream of SUMM2 (Fig. 3). We also show that LET2/MDS1 interacts with MEKK2 and SUMM2 and its stability is regulated by MEKK2 (Fig. 4a–e). Thus, LET1 and LET2/MDS1 might have similar and additive function in modulating *mekk1-mkk1/2-mpk4* autoimmunity. This is supported by the observation that the *let1/2* double mutant further alleviated *mekk1, mkk1/2*, and *mpk4* autoimmunity compared to *let1* or *let2* single mutants (Fig. 2). Notably, the single mutants of *let1* or *let2* clearly suppressed *mekk1-mkk1/2-mpk4* cell death (Fig. 2), suggesting

that LET1 and LET2/MDS1 might not simply function redundantly in regulating SUMM2 activation. Indeed, we observed that LET1 interacts with LET2/MDS1, and importantly, expression of LET2/MDS1 promotes LET1 phosphorylation (Fig. 4). Consistent with this observation, mutations of either *LET1* or *LET2/MDS1* in the *let1* or *let2* single mutants lead to the inactivation of SUMM2 in *mekk1-mkk1/2-mpk4* cell death. The additive effect of LET1 and LET2/MDS1 in regulating *mekk1-mkk1/2-mpk4* cell death also suggests that LET1 and LET2/MDS1 might have independent functions in this pathway. This could be due to that LET1 and LET2 might also form LET1 or LET2/MDS1 homodimers, in addition to LET1/2 heterodimer. In addition, LET1 and LET2/MDS1 might be activated by different ligands in modulating *mekk1-mkk1/2-mpk4* cell death.

We have shown that MEKK2 likely plays a structural role, rather than functions as a kinase, in regulating SUMM2 activation[52]. Consistently, MEKK2 scaffolds LET1 and SUMM2 for signaling activation[49]. However, both LET1 and LET2/MDS1 have autophosphorylation activity, and their kinase activity is required for their functions (Fig. 1g–i)[49]. Thus, LET1 and LET2/MDS1 are authentic kinases in the activation of SUMM2. It has been proposed that NLRs are kept in an inactive form by intra-molecular interaction (such as interaction between NBS and LRR domains), and disruption of intramolecular interaction activates NLRs[6,8]. LET1/2 may activate SUMM2 through a phosphorylation-based conformational change of SUMM2 to disrupt its intramolecular interaction. LET1/2 phosphorylation may also induce oligomerization of SUMM2 for NLR activation[53]. It has been proposed that CRCK3, a MPK4 substrate, is guarded by SUMM2 to monitor the integrity of the MEKK1-MKK1/2-MPK4 cascade[48]. It will be interesting to determine whether there is a connection between LET1/2 and CRCK3-mediated phosphorylation in the activation of SUMM2.

The GPI-anchored proteins LRE and LLGs have been proposed as co-receptors of FER, BUSP1/2 and ANX1/2, in regulating plant growth, reproduction, and immunity[28–31]. Our VIGS screen indicates that LLG1, but not LLG2, LLG3, nor LRE, regulates *mekk1*, *mkk1/2*, and *mpk4* autoimmunity (Figs. 5 and 6). This is consistent with the observation that *LLG1* is expressed in seedlings, whereas *LRE* is a female gametophyte-expressed gene[54]. LLG2 and LLG3 are strongly expressed in pollens and regulate pollen cell wall integrity[30]. Epistasis analysis indicates that, similar with LET1/2, LLG1 functions downstream of MEKK2 and upstream of SUMM2 in the *mekk1-mkk1/2-mpk4* cell death pathway (Fig. 6 and Supplementary Fig. 7). LLG1 interacts with the ectodomain of LET1/2 and mediates LET1/2 transport to the plasma membrane (Fig. 7). Thus, LRE and LLGs function as shared co-receptors of different *Cr*RLK1Ls, and their function specificity is determined by their spatiotemporal expression pattern. Interestingly, LLG1 also associates with PRR complex, contributing to the accumulation of PRR FLS2 and regulating plant immunity[32]. This suggests that, in addition to *Cr*RLK1Ls, LRR-RLKs could also be regulated by LRE and LLGs. However, it remains unknown whether LLG1 functions in plant immunity through an independent pathway, or through interaction with *Cr*RLK1Ls, such as FER and ANX, both of which have been shown to regulate plant immunity via modulating PRR complexes[24,25]. Emerging evidence indicates the extracellular peptides of the RALF family act as the ligands of *Cr*RLK1Ls[22,24,29,55,56]. RALFs or other type of ligands could be the potential ligands of LET1/2-LLG1 module in regulating SUMM2 activation.

Altogether, our results reveal that two *Cr*RLK1Ls, LET1 and LET2/MDS1, together with GPI-anchored protein LLG1, form a trimeric complex to modulate NLR SUMM2, which is activated in the absence of MEKK1-MKK1/2-MPK4 cascade. LLG1 likely functions as a co-receptor of LET1/2 and assists LET1/2 protein maturation and delivery from endoplasmic reticulum (ER), Golgi to plasma membrane (PM) (Fig. 7g).

## Methods

**Quantification and statistical analysis.** Data for quantification analyses are presented as mean ± standard error (SE) or standard deviation (SD). The different letters indicate the significant difference determined by one-way ANOVA followed by the Tukey test ($P < 0.05$). Number of replicates is shown in the figure legends.

**Plant materials.** The *Arabidopsis thaliana* ecotype Col-0 was used as wild type (WT). The T-DNA insertion lines, *SALK_139579* (*let2-1*, AT5G38990), *SALK_066322* (*let2-2*, AT5G38990), *SALK_074670C* (*mds3-1*, AT5G39020), *SALK_007613C* (*mds4-1*, AT5G39030), *SALK_029056C* (AT3G51550), *SALK_133057C* (*anx2-2*, AT5G28680), *SALK_016179C* (*anx1-1*, AT3G04690), *SALK_105055C* (*herk2*, AT1G30570), *SALK_083442C* (*cap1-1*, AT5G61350), *SALK_018797C* (*curvy1*, AT2G39360), *SALK_114667C* (*anj-1*, AT5G59700), *SALK_008043C* (*herk1-1*, AT3G46290), *SALK_007108* (*mds2-1*, AT5G39000), *SAIL_907_G02* (AT5G24010), *SAIL_809_D01* (AT5G24010), *SALK_033062* (AT4G39110), and *SAIL_448_D02* (AT2G21480), of different *Cr*RLK1L family members, *SAIL_47_G04* (*llg1-1*) for LLG1, *SALK_040289* (*ire-3*) and *CS66103* (*ire-6*) for LORELEI, *SALK_018793C* (*let1-1*), *SALK_052557* (*mekk1*), and *SALK_150039C* (*mekk2*) were ordered from Arabidopsis Biological Resource Center (ABRC). The seeds of *herk1-1the1-4* were obtained from Dr. Yanhai Yin[57]. The seeds of *fer-4* (CS69044), *llg1-2* (*SALK_086036*) and *pLLG1::HA-LLG1/llg1-2* were obtained from Dr. Alice Cheung[28]. The seeds of *llg1-3* were obtained from Dr. Dingzhong Tang[32]. The seeds of *llg2-1*, *llg3-1*, and *llg2-1llg3-1* were obtained from Dr. Lijia Qu[30]. The seeds of *mkk1/2* were obtained from Dr. Patrick Krysan[47]. The seeds of CRISPR/Cas9-generated double, triple, and quadruple *mds* mutants were previously reported[50]. The genotype of all the mutants was confirmed with PCR using the primers listed in Supplementary Table 1.

**Growth conditions.** Plants were grown in the growth rooms with 22 °C, 50–60% relative humidity, 70 μE m$^{-2}$ s$^{-1}$ light under 10/14 h light/dark cycles, except where indicated. The seedlings were grown on ½MS medium plates supplemented with 0.5% sucrose, 0.8% agar, and 2.5 mM MES at pH 5.7. The seeds were cold treated for two days at 4 °C before moving to a growth room. For investigating recovery of cell death at high temperature, the seedlings were grown on a ½MS plates in a 22 °C growth room with for 3 days after cold treatment, and then transferred to a 28 °C growth room with the indicated time.

**Plasmid constructs.** The constructs of *pTRV–RNA1* and *pTRV–RNA2* of *pYL156-GFP* and *pYL156-MEKK1* have been reported[51]. The DNA sequence of *LET2/MDS1* contains the restriction enzyme sites of BamHI and StuI, which are commonly used in our plant expression and binary vectors. The *LET2/MDS1* cDNA fragment containing NcoI-BglII at 5′ and SmaI-SnaBI at 3′ was amplified by PCR and digested by NcoI and SnaBI. The digested fragment was cloned into the linearized *pHBT-HA* vector digested by NcoI and StuI to get the intermediate construct of *pHBT-LET2-HA*. The fragment of *LET2/MDS1* digested by BglII and SmaI from the *pHBT-LET2-HA* were cloned into the linearized vectors of *pHBT-HA*, *pHBT-FLAG*, *pHBT-GFP*, *pCB302-HA*, and *pHBT-mCherry* digested by BamHI and StuI or SmaI to generate the constructs of *pHBT-LET2-HA*, *pHBT-LET2-FLAG*, *pHBT-LET2-GFP*, *pCB302-LET2-HA*, and *pHBT-LET2-mCherry*. The *pHBT-LET2$^{KM}$-HA* was generated by Platinum *Pfx* DNA polymerase-mediated site-directed mutagenesis with *pHBT-LET2-HA* as a template. The fragment of *LET2$^{KM}$* digested by BglII and SmaI from the *pHBT- LET2$^{KM}$-HA* was cloned into linearized vectors of *pCB302-HA* digested by BamHI and StuI to generate the construct of *pCB302- LET2$^{KM}$-HA*.

The *LET1* (AT2G23200) gene (2502 bp) was amplified by PCR from Col-0 cDNA using the primers containing BamHI at the 5′ end and StuI at the 3′ end[49]. Due to *LET1* fragment containing an internal BamHI site, the *LET1* PCR products were digested by BamHI and StuI into two fragments, *LET1$^N$* (BamHI-*LET1$_{1-1969 bp}$*-BamHI) and *LET1$^C$* (BamHI-*LET1$_{1970-2502 bp}$*-StuI). *LET1$^C$* was firstly cloned into linearized *pHBT-HA* digested by BamHI and StuI, and subsequently, *LET1$^N$* was introduced by BamHI digestion and ligation to obtain p*HBT-LET1-HA*[49]. To sub-clone *LET1* into other vectors by BamHI and StuI, site-directed mutagenesis by Platinum *Pfx* DNA polymerase-mediated PCR was used to mutate the internal BamHI site without changing its codons in *LET1* and generate *pHBT- LET1$_{mBamHI}$-HA*. Then *LET1$_{mBamHI}$* were sub-cloned into HBT-HA, HBT-GFP, and pCB302-HA vector through BamHI and StuI. The fragments of *MPK4* (AT4G01370, 1128 bp), *MEKK2* (AT4G08480, 2319 bp), and *SUMM2* (AT1G12280, 2682 bp) were amplified from Col-0 cDNA using the primers containing BamHI at the 5′ end and StuI at the 3′ end, and ligated into *pHBT-FLAG* vector[49]. *LET1$^{KM}$* and *SUMM2$^{ac}$* were generated by site-directed mutagenesis using Platinum *Pfx* DNA polymerase-mediated PCR. *MEKK2* and *SUMM2$^{ac}$* were sub-cloned into the binary vectors *pMDC32-2x35S::HA* or *pMDC32-2x35S::GFP* by BamHI and StuI digestions. The fragments of extracellular malectin-like domain (ECD) of *LET1* (LET1$^{ECD}$, 1-400aa) and the extracellular juxtamembrane (exJM) of *LET1* (LET1$^{exJM}$,

338-400aa) were amplified by PCR and cloned into the vector of *pHBT-HA* or a modified Glutathione S-transferase (GST) fusion protein expression vector *pGSTu* by BamHI and StuI digestion to generate the constructs of *pHBT-LET1^ECD^-HA* and *pGSTu-LET1^exJM^*. The *p35S::HA-LLG1* construct in plant expression vector was obtained from Dr. Alice Cheung[28]. The fragment of *LLG1* without signal peptide (△SP, 25-168aa) was amplified by PCR and digested by BglII and PstI, then ligated with a linearized maltose-binding protein (MBP) fusion protein expression vector *pMAL* (New England BioLabs, USA) by BamHI and PstI digestion to generate *pMAL-LLG1^△SP^*. *LET1^CD^* (1273-2502 bp) was amplified by PCR from *pHBT-35S:: LET1-HA* using the primers containing StuI at the 5′ end and KpnI at the 3′ end and cloned into an insect cell expression vector *pAcGHLT-C* to generate *pAcGHLT-LET1^CD^*[49] *LET2^ex^* (64-1320 bp) and *LET2^CD^* (1390-2640 bp) were amplified by PCR from *pHBT-35S::LET2-HA* using the primers containing BamHI at the 5′ end and HindIII at the 3′ end and cloned into *pET28* to generate *pET28- LET2^ex^* and *pET28- LET2^CD^*.

All the primer sequences are listed in Supplemental Table 1. The sequences of all genes and mutation were verified by the Sanger-sequencing. The binary plasmids were transformed into *Agrobacterium tumefaciens* strain GV3101 and introduced into *Arabidopsis* using the floral dipping method. Transgenic plants were selected by Glufosinate-ammonium (Basta, 50 μg/mL) for the *pCB302* vector and hygromycin (50 μg/mL) for the *pMDC32* vector. Multiple transgenic lines were analyzed by immunoblot (IB) for protein expression.

**Agrobacterium-mediated virus-induced gene silencing assay**. The binary TRV vector *pTRV-RNA1* and *pTRV-RNA2* derivatives, *pTRV-MEKK1*, *pTRV-CLA1*, and *pTRV-GFP* (the vector control), were transferred into *A. tumefaciens* strain GV3101 by electroporation. Positive transformants were selected on LB plates containing 50 μg/mL kanamycin and 25 μg/mL gentamicin by incubating at 28 °C for 36 h. An individual transformant was transferred into 2 mL LB liquid medium containing 50 μg/mL kanamycin and 25 μg/mL gentamicin in 20 mL glass culture tubes for overnight at 28 °C in a roller drum, and sub-cultured in 100 times of volume of fresh LB liquid medium containing 50 μg/mL kanamycin, 25 μg/mL gentamicin, 10 mM MES, and 20 μM acetosyringone for overnight at 28 °C with 200 rpm shaking. Cells were pelleted by 1300 g centrifugation, re-suspended in buffer containing 10 mM MgCl₂, 10 mM MES, and 200 μM acetosyringone, adjusted to OD₆₀₀ of 1.5 and incubated at 25 °C for at least 3 h. Bacterial cultures containing *pTRV-RNA1* and *pTRV-RNA2* derivatives were mixed at a 1:1 ratio and inoculated into the first pair of true leaves of 2-week-old soil-grown plants using a needleless syringe.

**Transient expression in *Arabidopsis* protoplasts**. The indicated *pHBT* constructs were used for protoplast transfection following the protocol[58]. Briefly, for Co-IP assay, 100 μL of plasmid DNA (2 μg/μL) was mixed with 1 mL of protoplasts (2 × 10⁵ cells/mL) for the PEG-mediated transfection.

**Co-immunoprecipitation assay**. Proteins were expressed overnight in *Arabidopsis* protoplasts or *N. benthamiana* leaves for 3 days. Protoplasts were lysed by vortexing and leaves were grounded in the extraction buffer (100 mM NaCl, 1 mM EDTA, 10 mM HEPES, pH 7.5, 2 mM NaF, 2 mM Na₃VO₄, 1 mM DTT, 0.5% Triton X-100, 10% glycerol, and 1 x protease inhibitor). After centrifugation at 12,500×g at 4 °C for 15 min, 250 μL of extraction buffer were added to dissolve pellets, and 20 μL of supernatant were collected for input controls, and the remaining was incubated with α-FLAG affinity beads (Sigma, USA) at 4 °C for 2 h with gentle shaking. Beads were collected and washed three times with washing buffer (10 mM HEPES, pH 7.5, 100 mM NaCl, 1 mM EDTA, 1% Triton X-100), and once with 50 mM Tris-HCl, pH 7.5. Proteins were eluted by 2 × SDS-PAGE loading buffer and boiled at 94 °C for 5 min. Immunoprecipitated and input proteins were analyzed by immunoblot with indicated antibodies.

**Trypan blue and DAB staining**. For investigating cell death and H₂O₂ accumulation in leaves, the cotyledons or leaves were detached and soaked into 2.5 mg/mL trypan blue solution (the powder of trypan blue was dissolved in lactophenol with 1:1:1:1 ratio of lactic acid, glycerol, liquid phenol, and ddH₂O) or DAB solution (1 mg/mL DAB dissolved in ddH₂O, pH 3.8) for overnight incubation. Samples were then destained by trypan blue destaining solution (the mixture of lactophenol and ethanol with 1:2 ratio) or DAB destaining solution (the mixture of glycerol, acetic acid and ethanol with 1:1:3 ratio) respectively with gentle shaking at 70 rpm at room temperature for 1 day. The samples were observed and recorded under a dissecting microscope.

**RNA isolation and qRT-PCR analysis**. Plant total RNAs were extracted by TRIzol reagent (Sigma/Invitrogen, USA). Genomic DNA was degraded by treatment with RNase-free DNase I (NEB, USA). Complementary DNAs (cDNAs) were synthesized with M-MuLV Reverse Transcriptase (NEB, USA) and oligo(dT) primers. Quantitative RT-PCR analysis was performed by iTaq Universal SYBR green Supermix (Bio-Rad, USA) with a Bio-Rad CFX384 Real-Time PCR System (Bio-Rad, USA). UBQ10 was used as an internal reference.

**Recombinant protein isolation from *E. coli* and in vitro pull-down assay**. Fusion proteins were produced from *E. coli* BL21 at 16 °C using LB medium with 0.25 mM isopropyl β-D-1-thiogalactopyranoside (IPTG). HIS-SUMO-LET2^ex^ and HIS-SUMO-LET2^CD^ were purified with Ni-NTA agarose (Qiagen, USA). GST and GST-LET1^exJM^ fusion proteins were purified with Pierce glutathione agarose (Thermo Scientific, USA), and MBP-LLG1 fusion proteins were purified using amylose resin (New England Biolabs, USA) according to the standard protocol from companies. MBP-LLG1 proteins were incubated with GST or GST-LET1^exJM^ in the pull-down buffer (20 mM Tris-HCl, pH 7.5, 100 mM NaCl, 0.1 mM EDTA, 0.2% Triton X-100) for 1 h with gentle shaking, subsequently incubated with 20 μL of glutathione agarose beads at 4 °C for another 2 h with gentle shaking. Beads were washed five times with pull-down buffer (20 mM Tris-HCl, pH 7.5, 100 mM NaCl, 0.1 mM EDTA, and 0.2% Triton X-100). Beads were boiled in 50 μL of 2x SDS protein loading buffer for 10 min and detected by immunoblotting with an α-MBP or α-GST antibody. For HIS fusion protein pull-down assay, about 10 μg of HIS-SUMO-LET2^ex^ or HIS-SUMO-LET2^CD^ proteins were mixed with the LET1-FLAG cell lysates in the IP buffer (10 mM HEPES, pH 7.5, 100 mM NaCl, 10% glycerol, and 0.5% Triton X-100) at 4 °C for 30 min with gentle shaking, subsequently incubated with 20 μL of Ni-NTA agarose (Qiagen, USA) at 4 °C for another 30 min with gentle shaking. The beads were harvested by centrifugation and washed five times with the IP buffer and one time with washing buffer (20 mM Tris, pH 8.0, 500 mM NaCl, and 10 mM imidazol). The pull-down proteins were eluted by 50 μL of elution buffer (20 mM Tris, pH 8.0, 150 mM NaCl, and 250 mM imidazole) and detected by an immunoblot with an α-FLAG antibody.

**In vitro kinase assay**. The in vitro kinase assays were performed with 0.5 μg fusion proteins of GST, HIS-GST-LET1^CD^ or HIS-SUMO-LET2^CD^ in 20 μL of kinase reaction buffer (20 mM Tris-HCl, pH 7.5, 10 mM MgCl₂, 5 mM EGTA, 100 mM NaCl, 1 mM DTT, and 1 μL [γ-32P] ATP). After gentle shaking at room temperature for 2 hr, samples were denatured with 4x SDS loading buffer and separated by 10% SDS-PAGE gel. Phosphorylation was analyzed by autoradiography. For immuno-complex kinase assay, GFP-FLAG, LET1-FLAG, or LET1^KM^-FLAG were transiently co-expressed with LET2-HA or LET2^KM^-HA in protoplasts for 10 h, and purified by α-FLAG agarose. The proteins were incubated with 20 μL of kinase reaction buffer at room temperature for 3 hr with gentle shaking. The reactions were stopped by adding 4× SDS protein loading buffer. The phosphorylation of proteins was analyzed by autoradiography after separation with 10% SDS-PAG.

**Confocal Microscopy and FLIM-FRET assays**. The GFP and mCherry fusion proteins were detected using a Leica TCS SP8 confocal laser scanning microscope (Germany). The GFP fluorescence was excited at 488 nm, and emissions were detected between 490 and 530 nm. The mCherry fluorescence was excited at 587 nm, and emissions were detected between 590 and 620 nm. The pinhole was set at 1 Airy unit. Images and FLIM/FRET analyses were performed by using Leica Application Suite X (LAS X) software as described[59]. Briefly, FRET measurements were done with a pair of GFP/mCherry fusion proteins. The image of GFP donor fluorescence was analyzed and scanned at 488 nm and detected between 490 and 530 nm. The fluorescence lifetime (τ) was calculated as the average of 20τ values randomly measured in the protoplast cells. The values obtained for 15 protoplasts were used to determine the average value of τ for each pair of proteins analyzed. The relative fluorescence intensity (I) in a certain region of interest (ROI), lifetime (τ) and FRET efficiency were measured by the Leica LAS X software. FRET efficiency (*E*) was calculated by using the formula $E = 1-(\tau_{DA}/\tau_D)$, $\tau_{DA}$ is the lifetimes of the donor in the presence of acceptor and $\tau_D$ is fluorescence lifetime of the donor alone.

**Reporting summary**. Further information on research design is available in the Nature Research Reporting Summary linked to this article.

## Data availability
The source data for Figs. 1 and 3–7 and Supplementary Figs. 3 and 6–7 are provided as a Source Data File. Other original data that support the findings of this study are available from the corresponding author upon request. Source data are provided with this paper.

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

## Acknowledgements

We thank the *Arabidopsis* Biological Resource Center (ABRC) for *Arabidopsis* T-DNA insertion library and various mutant seeds, Drs. Alice Cheung (University of Massachusetts, Amherst, USA), Patrick Krysan (University of Wisconsin, Madison, USA), Yanhai Yin (Iowa State University), Dingzhong Tang (Fujian Agriculture and Forestry University, China), Lijia Qu (Peking University, China), and Yuelin Zhang (University of British Columbia, Canada) for *Arabidopsis* seeds and constructs, Tanja Pfeiler, Doris

Keck and Peter Stasnik for genotyping the *MDS* CRISPR/Cas mutants, and members of the laboratories of L.S. and P.H. for discussions and comments of the experiments. The work was supported by National Institutes of Health (NIH) (R01GM092893) and National Science Foundation (NSF) (MCB-1906060) to P.H., NIH (R01GM097247) and the Robert A. Welch Foundation (A-1795) to L.S, PEW Latin American Fellows Program to F.A.O.M., and the Austrian Science Fund project FWF I 1725-B16 to M.T.H. Y.H., C.Y., and D.G. were partially supported by China Scholarship Council (CSC).

## Author contributions

Y.H., C.Y., J.L., L.S., and P.H. conceived the project, designed experiments and analyzed data. Y.H., C.Y., J.L., B.F., D.G., L.K. and F.A.O.M. performed experiments and analyzed data. J.R., and M.T.H. generated *mds* CRISPR/Cas lines. W.M.W. analyzed data and provided critical feedback. Y.H., L.S., and P.H. wrote the manuscript with inputs from all co-authors.

## Competing interests

The authors declare no competing interests.
