## [Peer Review File · Nature Communications]

Reviewers' comments:

Reviewer #1 (Remarks to the Author):

I think this is a very well-executed study that adds an important and interesting regulatory node to the already very elaborate-regulated MeKK1-regulated cell death/growth pathway.

I only have three minor queries (1,2,3), they are embedded in the review.

This manuscript exhaustively demonstrates that of the cluster of 4 closely related CrRLKs, only one of them MDS1 (which the authors termed LET2) suppressed mekk1-induced cell death. LET2 was named based on a selection strategy developed by the authors' group to identify lethality suppressor of mekk1, which has been reported and successfully used before. This study describes a novel signaling complex involving LET2 and a homolog LET1 that together formed a trimeric complex to regulate the MEKK1-controlled cell death path, a system that the authors have extensively studied also. The identification and functional linkage between the LET2-MEKK linkage was established through a series of expertly executed and well-established approaches.

Fig. 1-4 report a series of detailed genetic and phenotypic (from cell death, plant growth, loss and gain of function analyses) to establish the functional relationship between LET1/2 and MEKK1/2 and SUMM2, a signaling component downstream of MEKK2. They showed that LET2 acts downstream of MEKK2 and upstream of SUMM2. The experiments were skillfully carried out and augmented the understanding of the MEKK1/2 regulated cell death/growth pathway with a crucial player. They also demonstrated that LET2 regulates LET1 phosphorylation.

Fig. 5 onwards are genetics and biochemical experiments that demonstrate how the LET1/2 complex acts on the MEKK1 system and led to the discovery of the LET1/2-LLG1 trimeric complex as a functional unit, and established that LLG1 functions comparably as LET1/2 in its capacity as regulator of MEKK1 based on their original VIGS-silenced MEKK1 system. The authors tested all the other members of the LLG family and ruled out their involvement, consistent with the relatively high expression specificity in reproductive tissues and cells. Interesting they also tested llg1-3, a point mutation that led to the discovery of LLG1 as critical for immunity responses. llg1-3, unlike llg1-1 and 1-2, which are T-DNA induced nulls, did not have growth defect, nevertheless comparably suppressed RNA-induced MEKK1-induced growth retardation.

I could have missed it, but I have two questions (1) how would authors interpret this? and (2) in light of the later results in experiments carried out in loss of function mekk1 mutants (Fig. 5), did they test llg1-3 in that background?

However, studies based on combining loss of LLG1 and loss of MEKK1 revealed additive effect on mekk1-induced almost lethal, hinting to loss of LLG1 most likely had acted via potentially its interaction with the LET RKs, based on a previous finding that relates LLG1's function as a chaperone for the LET homolog FERONIA. To pursue these, they discovered that in fact loss of LLG1 had a fundamental impact on the signaling capacity of LET1/2 in that their secretion to the cell surface was prohibited, thus leading to the aggravated growth phenotype. In these studies, they also identified that FERONIA also acts in a similar capacity as LET1/2 in that loss of FER also severely aggravated mekk1's severe growth defects.

This led to the question (3) of are FER and LET1/2 only, or there may be others in the CrRLKs are also similarly engaged, e.g., given that the MDIs were the closest homologs to LET2/1, I wonder if the authors have tested the other MDISs in mekk mutant background to look for their contribution, even though loss of these genes was shown not to be suppressive of RNAi-MEKK1 earlier on.

Reviewer #2 (Remarks to the Author):

In this manuscript, Huang et al. identified a trimeric CrRLK1L module that regulates the activation of the NLR-mediated autoimmunity. This trimeric CrRLK1L module consists of the CrRLK1L type of receptor-like kinases LET1, LET2 and the GPI anchored LLG1. The authors showed that LET2 and LLG1 regulate mekk1-mkk1/2-mpk4 cell death and SUMM2 activation. And LLG1 interacts with LET1/2 and mediates LET1/2 transport to the plasma membrane, acting as a co-receptor of LET1/2 in the mekk1- mkk1/2-mpk4 cell death pathway. This work connects the immune function of MAPK cascade and the GPI anchored protein with RLKs and NLR. Giving the important roles of MAPK cascade and GPI anchored protein LLG1 in many biological processes, this work will be highly appreciated by readers from different fields in plant biology. The study is thorough, and the data are convincing and very well presented. I only have some minor points:

1) Fig 4B, The authors showed that MEKK2-GFP affected LET2-HA accumulation in protoplast assay. To be more convincing, please add a control, for instance, using the GTP alone or GFP fused with other gene as a control, and show it does not affect LET2 accumulation.

2) The let1 mutation has stronger effects than that of let2 on mekk2 phenotypes, and let1 let2 double has even stronger effects than each of the single mutation. Those observations indicated that LET1 and LET2 may have independent roles. Please discuss the potential mechanism on how LET1 and LET2 might function independently in the mekk2-mediated cell death. Could LET1 and LET2 function alone when the trimeric CrRLK1L-LLG1 complex is not formed appropriately in the absence of the other LET protein?

3) The authors proposed a model in Fig 7, and only explained the model in the figure legends. However, it would be helpful if the authors could briefly describe the model in the main text.

4) Fig 1G, Please define RBC in the figure legends

5) P12, Subheading, "The llg1-1 mutant suppresses mkk1/2 and mpk4 cell death." I would suggest that the authors change the "mutant" to "mutation", which I think is more accurate.

6) Fig 3E, Legends, The authors stated that "68 and 71 independent primary (T1) transgenic plants carrying 35S::SUMM2ac-HA in WT and let2-1 were characterized". However, as indicated in the main text, only around half of them showed growth defects, cell death, H2O2 accumulation and elevated expression of PR genes. To be accurate, the authors should indicate that the two representative plants, which showed the growth defects, were shown in the Figure legends.

7) Fig 4F, Please explain what is "I" at the left of the upper panel? Was it the mCherry alone?

Reviewer #3 (Remarks to the Author):

In this study, a genetic approach was employed to identify genes involved in signaling pathways downstream of the MAP3K MEKK1 and leading to the activation of cell death. Out of a list of candidate genes representing CrRLK1L family members, LET2/MDS1 was identified as required for activation of cell death caused by VIGS of MEKK1 in Arabidopsis. Genetic analysis using let1, let2, mekk1, mkk1/2, and mpk4 mutant combinations demonstrated that LET1 and LET2 are required additively for the activation of cell death by a non-functional MEKK1-MEK1/2-MPK4 signaling cascade. In addition, genetic evidence indicated that LET2/MDS1 function downstream of MEKK2. Biochemical analysis revealed interactions between LET2/MDS1 and MEKK2 (in agreement with the genetic data), SUMM2 and LET1, but not with MPK4. Interestingly, LET1 was found to be phosphorylated when coexpressed with LET2/MDS1 but not with LET2/MDS1 mutated in the ATP binding site and putatively kinase deficient. Additional genetic analysis identified another gene

encoding the LLG1 protein that is required for the activation of cell death by a non-functional MEKK1-MEK1/2-MPK4 signaling cascade. Epistatic analysis indicated that LLG1 acts downstream of MEKK2 and independently of SUMM2. Biochemical assays demonstrated that LLG1 interacts with both LET1 and LET2/MDS1 and is important for their transport from the ER to the plasma membrane. The genetic data presented here are original, very solid and supported by a detailed documentation. They add new players in a very interesting MAP kinase pathway that negatively regulates NLR-mediated autoimmunity. Less insightful are the biochemical data, which remain preliminary and need further validation. The interpretation of the results is too speculative and the molecular mechanisms proposed are not sufficiently elaborated and supported by experimental data.

Title:

-The main findings derive from genetic interactions and not from biochemical studies and this is not reflected in the title.

-In the Title: "A trimeric complex regulates..."; in the Abstract: "We have identified a trimeric complex..that senses the disturbance of immune signaling and regulates the activation of NLR SUMM2 for initiating cell death and autoimmunity".

No direct evidence is shown for a complex including LET1, LET2/MDS1, and LLG1, for its ability to sense disturbances in immune signaling and to regulate signaling pathways.

Introduction:

"LET1 and LET2 hetero-dimerize.."

According to the experimental data shown, LET1 and LET2 coimmunoprecipitate (Fig. 4E) and are in close proximity as observed by FRET experiments (Fig. 4F-G). This is sufficient to support the notion that they interact either directly or indirectly. Whether they are part of a multiprotein complex, a dimer or a trimer it remains to be established.

"LET2 promotes LET1 phosphorylation suggesting a phosphoregulation between different CrRLK1Ls". The evidence for this claim is circumstantial and not supported by follow up analysis. Is LET2 a functional kinase? Does it phosphorylates LET1 in vitro? Is LET1 and LET2 kinase activity and/or phosphorylation modulated in vivo by activation of cell death (e.g. MEKK1 silencing)? The evidence that LET1 is phosphorylated when coexpressed with LET2 and that a mutation in the LET2 ATP binding site make LET2 non-functional is interesting, but it is not sufficient to provide an activation mechanism for LET1 by LET2/MDS1.

"Thus, a specific trimeric module consisting of LET1, LET2 and GPI-anchored LLG1 senses the disturbance caused by a deficient MEKK1-MKK1/2-MPK4 cascade, and modulates SUMM2-mediated autoimmunity." As mentioned above, there is no biochemical or other evidence that LET1, LET2 or LLG1 senses the effect of an inactive MEKK1-MKK1/2-MPK4 cascade. There is also no proof that they form a trimeric module and regulate SUMM2 activity.

Results:

-The rationale and minimal background information for the various experiments is often missing making hard to understand the experimental design and the hypotheses behind it.

-Page 5 paragraph: "The mutations in.."

It is not clear at a first sight why LET1 is excluded from the targeted screen. It should be clearly mentioned that a mutation in LET1 was found in a previous screen to inhibit the cell death induced by MEKK1 silencing. This knowledge also provides the rationale to test other family members.

-Figure 1. LET2/MDS1 should be consistently indicated with both names in all parts of the manuscript (including Fig. 1D).

-Page 6: "Thus the data support that only LET2/MDS1... is involved in the regulation of mekk1 cell

death." Here and in many other parts of the manuscript, the term "regulation" is inappropriately used. The genetic experiments presented here indicate that LET2/MDS1 plays a role in the activation of cell death caused by MEKK1 silencing. No regulatory role is demonstrated for LET2/MDS1.

-Page 7: "The let2mekk1 mutant was slightly smaller than let1mekk1..."

It is the first time that this mutant is mentioned in the text. The comparison is not obvious and thereon why the let1mekk1 mutant is used should be explained.

-Page 8: "Since LET2/MDS1 functions in the same pathway with MEKK2 for SUMM2 activation..."

This claim at the present stage of the manuscript is not supported by results shown so far. Up to this point, the only evidence related to the possibility that LET2/MDS1 is in the same pathway of MEKK2 is the observation that the phenotype of the mekk2let1/2mpk4 quadruple mutant is similar to that of the let1/2mpk4 or mekk2mpk4. This is not sufficient to conclude that LET2/MDS1 is involved in SUMM2 activation through MEKK2.

-Page 9: "Taken together, LET2/MDS1 functions genetically downstream of MEKK2 and upstream of SUMM2 in modulating the mekk1-mkk1/2-mpk4 cell death pathway". Again, overinterpretation of the results: the evidence that overexpression of an active SUMM2 variant causes a similar phenotype in let2 mutant and wt plants is not sufficient to claim that LET2/MDS1 functions upstream of SUMM2. Also, it appears that the mekk1-mkk1/2-mpk4 cell death pathway modulates LET2/MDS1 and MEKK2, and not the opposite.

-Fig. 4B: The data presented showing that LET2 accumulation is increased by coexpression of MEKK2 is not enough reliable: appropriate controls are missing in this experiment: 1. Coexpression of LET-HA and MEKK2-GFP should be compared to coexpression of LET-HA and GFP; 2. The Coomassie blue staining of Rubisco is not sufficient to demonstrate equal loading of samples, particularly in an experiment where protein accumulation is monitored. More importantly, in protoplasts there is no effect of MEKK2-FLAG expression on LET2-HA accumulation (Fig. 4A): in the input panels of the figure, there is no difference in the accumulation of LET2-HA in the presence or absence of MEKK2-FLAG, lanes 1 and 2.

-Fig. 4C-D: The observation that LET1 is phosphorylated when coexpressed with LET2/MDS1 is intriguing, but not sufficiently followed up for drawing solid conclusions about the role of LET2/MDS1 kinase activity in signaling: is LET2/MDS1 an active protein kinase? Is LET1 phosphorylated in vivo upon induction of cell death by an inactive MEKK1-MKK1/2-MPK4 cascade or by overexpression of MEKK2? Is this phosphorylation required for LET1 activation?

-Page 10 "We tested whether LRE/LLGs are involved in LET1/2-mediated mekk1 cell death..."
How many LREs and LLGs are encoded in the Arabidopsis genome? How many were tested? How single and double mutants included in this study were selected?

-Page 11. The authors point out that their observations related to the llg1 mutants are apparently contradicting: the llg1 mutants aggravated the growth defects caused by a mutation in mekk1, whereas they suppressed cell death triggered by VIGS of the MEKK1 gene. This contradiction is explained by the authors hypothesizing that "LGG1 plays one role in regulating initial seedling development in concert with MEKK1, and another role in regulating mekk1 cell death at later stages" and that "compared to genetic mutations, VIGS-mediated silencing bypasses the defects associated with embryonic and early seedling development."

If this is indeed the case, why subsequent experiments were carried out with mkk1/2 and mpk4 genetic mutants and not by silencing these genes by VIGS? Does MEKK1 signal through MKK1/2 and MPK4 during seedling development?

Discussion:

-Page 15: "In contrast, LET2/MDS1 share a similar function in regulating SUMM2 activation with

LET1." There is no evidence in this study for regulation of SUMM2 activation by LET2/MDS1 or LET1.

-Page 15: "We also show that LET2/MDS1 interacts with MEKK2 and SUMM2 and its stability is regulated by MEKK2". As indicated above, the evidence for regulation of LET2/MDS1 stability by MEKK2 is very weak and does not support this conclusion.

-Page 15: "The data indicate that LET2/MDS1 may function upstream of LET1 and phosphorylate LET1 for signaling activation." The data supporting this possibility are preliminary and limited to the observation that when the two proteins are transiently overexpressed in *N. benthamiana* plants LET1 is phosphorylated and this phosphorylation is only observed with LET2/MDS1 in the wild-type form but not with a putative kinase deficient mutant. This observation should be corroborated by other experiments before any conclusion can be drawn about the role of LET1 phosphorylation for signaling activation.

-Page 15: "Consistent with this observation, mutations of either LET1 or LET2/MDS1 in the *let1* or *let2* single mutants disrupt LET1/LET2 complex and signaling pathway, leading to inactivation of SUMM2." These data do not appear in the manuscript.

-The manuscript Yang et al. (submitted), is cited in the text but not provided.

Reviewers' comments:

Reviewer #1 (Remarks to the Author):

I think this is a very well-executed study that adds an important and interesting regulatory node to the already very elaborate-regulated MeKK1-regulated cell death/growth pathway.

I only have three minor queries (1,2,3), they are embedded in the review.

This manuscript exhaustively demonstrates that of the cluster of 4 closely related CrRLKs, only one of them MDS1 (which the authors termed LET2) suppressed mekk1-induced cell death. LET2 was named based on a selection strategy developed by the authors' group to identify lethality suppressor of mekk1, which has been reported and successfully used before. This study describes a novel signaling complex involving LET2 and a homolog LET1 that together formed a trimeric complex to regulate the MEKK1-controlled cell death path, a system that the authors have extensively studied also. The identification and functional linkage between the LET2-MEKK linkage was established through a series of expertly executed and well-established approaches.

Fig. 1-4 report a series of detailed genetic and phenotypic (from cell death, plant growth, loss and gain of function analyses) to establish the functional relationship between LET1/2 and MEKK1/2 and SUMM2, a signaling component downstream of MEKK2. They showed that LET2 acts downstream of MEKK2 and upstream of SUMM2. The experiments were skillfully carried out and augmented the understanding of the MEKK1/2 regulated cell death/growth pathway with a crucial player. They also demonstrated that LET2 regulates LET1 phosphorylation.

Fig. 5 onwards are genetics and biochemical experiments that demonstrate how the LET1/2 complex acts on the MEKK1 system and led to the discovery of the LET1/2-LLG1 trimeric complex as a functional unit, and established that LLG1 functions comparably as LET1/2 in its capacity as regulator of MEKK1 based on their original VIGS-silenced MEKK1 system. The authors tested all the other members of the LLG family and ruled out their involvement, consistent with the relatively high expression specificity in reproductive tissues and cells. Interesting they also tested llg1-3, a point mutation that led to the discovery of LLG1 as critical for immunity responses. llg1-3, unlike llg1-1 and 1-2, which are T-DNA induced nulls, did not have growth defect, nevertheless comparably suppressed RNA-induced MEKK1-induced growth retardation.

We thank this reviewer for the thorough summary and insightful comments of our work.

I could have missed it, but I have two questions (1) how would authors interpret this? and (2) in light of the later results in experiments carried out in loss of function *mekk1* mutants (Fig. 5), did they test *llg1-3* in that background?

Our response: 1). There are three *llg1* mutant alleles, *llg1-1*, *llg1-2* and *llg1-3*, that have been reported (Li et al., 2015; Shen et al., 2017). The T-DNA insertion mutants, *llg1-1* and *llg1-2*, have growth defects (Li et al., 2015), but *llg1-3* with a G114R mutation has a normal growth phenotype (Shen et al., 2017). To exclude the effect of growth defect of *llg1-1* and *llg1-2* on RNAi-*MEKK1* cell death (Fig. 5A-G), we silenced *MEKK1* in the *llg1-3* mutant (Fig. S4C and D). The *llg1-3* mutant also suppressed RNAi-*MEKK1* cell death, indicating that LLG1-regulated growth and MEKK1 cell death are uncoupled. We have clarified this in the revised text (Page 11, Paragraph 2).

2) Yes, we generated the *llg1-3 mekk1* double mutant which showed enhanced cell death similar with *llg1-1 mekk1* and *llg1-2 mekk1* as shown in Fig. S5A-C.

However, studies based on combining loss of LLG1 and loss of MEKK1 revealed additive effect on *mekk1*-induced almost lethal, hinting to loss of LLG1 most likely had acted via potentially its interaction with the LET RKs, based on a previous finding that relates LLG1's function as a chaperone for the LET homolog FERONIA. To pursue these, they discovered that in fact loss of LLG1 had a fundamental impact on the signaling capacity of LET1/2 in that their secretion to the cell surface was prohibited, thus leading to the aggravated growth phenotype. In these studies, they also identified that FERONIA also acts in a similar capacity as LET1/2 in that loss of FER also severely aggravated *mekk1*'s severe growth defects.

This led to the question (3) of are FER and LET1/2 only, or there may be others in the CrRLKs are also similarly engaged, e.g., given that the MDIs were the closest homologs to LET2/1, I wonder if the authors have tested the other MDISs in *mekk1* mutant background to look for their contribution, even though loss of these genes was shown not to be suppressive of RNAi-*MEKK1* earlier on.

Our response: We thank this reviewer's valuable suggestion. As this reviewer pointed out, we tested different MDS single and higher-order mutants in suppressing RNAi-*MEKK1* cell death by VIGS, and identified *let2/mds1*, not other *mds* mutants, as a suppressor, suggesting LET2/MDS1 is a major *CrRLK1L* gene involved in the modulation of *mekk1* cell death. We didn't make the genetic mutants of *mekk1* with other *mds* mutants since we did not observe their clear contribution on *mekk1* cell death in the initial VIGS screen. The reason we chose FERONIA as a potential candidate to test its effect on *mekk1* growth is because the *fer* mutants have similar growth defects with *llg1-1* and *llg1-2*. In addition, it has been shown that LLG1 acts as a chaperone and co-receptor of FER (Li et al., 2015; Xiao et al., 2019).

Reviewer #2 (Remarks to the Author):

In this manuscript, Huang et al. identified a trimeric CrRLK1L module that regulates the activation of the NLR-mediated autoimmunity. This trimeric CrRLK1L module consists of the CrRLK1L type of receptor-like kinases LET1, LET2 and the GPI anchored LLG1. The authors showed that LET2 and LLG1 regulate mekk1-mkk1/2-mpk4 cell death and SUMM2 activation. And LLG1 interacts with LET1/2 and mediates LET1/2 transport to the plasma membrane, acting as a co-receptor of LET1/2 in the mekk1- mkk1/2-mpk4 cell death pathway. This work connects the immune function of MAPK cascade and the GPI anchored protein with RLKs and NLR. Giving the important roles of MAPK cascade and GPI anchored protein LLG1 in many biological processes, this work will be highly appreciated by readers from different fields in plant biology. The study is thorough, and the data are convincing and very well presented.

We thank this reviewer for the comments and the recognition of our work.

I only have some minor points:

1) Fig 4B, The authors showed that MEKK2-GFP affected LET2-HA accumulation in protoplast assay. To be more convincing, please add a control, for instance, using the GFP alone or GFP fused with other gene as a control, and show it does not affect LET2 accumulation.

Our response: We thank this reviewer for the suggestion. We have included a control of GFP to show that GFP did not affect LET2/MDS1 protein accumulation in *Nicotiana benthamiana* (new Fig. 4B compare lane 1 and 2). In addition, we also included another control to show that MEKK2 did not affect GFP protein accumulation in *N. benthamiana* (new Fig. 4B compare lane 4 and 5).

2) The *let1* mutation has stronger effects than that of *let2* on mekk2 phenotypes, and *let1 let2* double has even stronger effects than each of the single mutation. Those observations indicated that LET1 and LET2 may have independent roles. Please discuss the potential mechanism on how LET1 and LET2 might function independently in the mekk2-mediated cell death. Could LET1 and LET2 function alone when the trimeric CrRLK1L-LLG1 complex is not formed appropriately in the absence of the other LET protein?

Our response: Based on our observation, the *let2*, *let1* and *let1let2* mutants have gradually increased effects on suppressing *mekk1-mkk1/2-mpk4* cell death, suggesting that LET1 and LET2/MDS1 could function together, and might also have independent roles in this pathway. We have shown that LET1 interacted with LET2/MDS1 to form a heterodimer, and promoted LET1 phosphorylation. The independent function of LET1 and LET2/MDS1 suggests that they might also form LET1 or LET2/MDS1 homodimers, in addition to LET1 and LET2 heterodimer. Furthermore, LET1 and LET2/MDS1 might be activated by different ligands in

modulating *mekk1-mkk1/2-mpk4* cell death. Since LET1 and LET2/MDS1 have additive effect on suppressing *mekk1-mkk1/2-mpk4* cell death, it is very likely that LET1 and LET2/MDS1 could function alone when the trimeric CrRLK1L-LLG1 complex is not formed appropriately in the absence of the other LET protein. We have added this in the Discussion (Page 16, paragraph 2)

3) The authors proposed a model in Fig 7, and only explained the model in the figure legends. However, it would be helpful if the authors could briefly describe the model in the main text.

Our response: We thank this reviewer for the suggestion, and have included the description of the model in the main text (page 17, last paragraph).

4) Fig 1G, Please define RBC in the figure legends

Our response: We have defined RBC and modified the sentence as “Coomassie Brilliant Blue (CBB) staining of RuBisCO (RBC) is shown as a loading control (lower panel).” in the figure legends.

5) P12, Subheading, “The *llg1-1* mutant suppresses *mkk1/2* and *mpk4* cell death.” I would suggest that the authors change the “mutant” to “mutation”, which I think is more accurate.

Our response: We thank this reviewer for the suggestion, and have made the change.

6) Fig 3E, Legends, The authors stated that “68 and 71 independent primary (T1) transgenic plants carrying 35S::SUMM2ac-HA in WT and *let2-1* were characterized”. However, as indicated in the main text, only around half of them showed growth defects, cell death, H₂O₂ accumulation and elevated expression of PR genes. To be accurate, the authors should indicate that the two representative plants, which showed the growth defects, were shown in the Figure legends.

Our response: We have made the change as the reviewer suggested. The new sentence now reads as “Two representative three-week-old plants, which showed the growth defects, and their controls, are shown in the figure.”.

7) Fig 4F, Please explain what is “I” at the left of the upper panel? Was it the mCherry alone?

Our response: We are sorry for the confusion. “I” is the vertical “-“, indicating no mCherry vector, only LET1-GFP, in that panel, which was served as a control. We have another control of BIR2-mCherry for interaction specificity. We have changed “I” to “Ctrl” to avoid the confusion.

Reviewer #3 (Remarks to the Author):

In this study, a genetic approach was employed to identify genes involved in signaling pathways downstream of the MAP3K MEKK1 and leading to the activation of cell death. Out of a list of candidate genes representing CrRLK1L family members, LET2/MDS1 was identified as required for activation of cell death caused by VIGS of MEKK1 in Arabidopsis. Genetic analysis using *let1*, *let2*, *mekk1*, *mkk1/2*, and *mpk4* mutant combinations demonstrated that LET1 and LET2 are required additively for the activation of cell death by a non-functional MEKK1-MEK1/2-MPK4 signaling cascade. In addition, genetic evidence indicated that LET2/MDS1 function downstream of MEKK2. Biochemical analysis revealed interactions between LET2/MDS1 and MEKK2 (in agreement with the genetic data), SUMM2 and LET1, but not with MPK4. Interestingly, LET1 was found to be phosphorylated when coexpressed with LET2/MDS1 but not with LET2/MDS1 mutated in the ATP binding site and putatively kinase deficient. Additional genetic analysis identified another gene encoding the LLG1 protein that is required for the activation of cell death by a non-functional MEKK1-MEK1/2-MPK4 signaling cascade. Epistatic analysis indicated that LLG1 acts downstream of MEKK2 and independently of SUMM2. Biochemical assays demonstrated that LLG1 interacts with both LET1 and LET2/MDS1 and is important for their transport from the ER to the plasma membrane. The genetic data presented here are original, very solid and supported by a detailed documentation. They add new players in a very interesting MAP kinase pathway that negatively regulates NLR-mediated autoimmunity.

We thank this reviewer for the comments and recognition of our work.

Less insightful are the biochemical data, which remain preliminary and need further validation. The interpretation of the results is too speculative and the molecular mechanisms proposed are not sufficiently elaborated and supported by experimental data.

During the revision, we have added new biochemical and molecular data to strengthen our conclusions, in particular LET2/MDS1 stabilization by MEKK2 (new Fig.4 B-E), LET2/MDS1 phosphorylation (new Fig. 1I, 4H), and LET2/MDS1 and LET1 complex formation (Fig. 4L). We also modified our interpretation of the results as suggested by the reviewer. Please see the detailed responses below.

Title:

-The main findings derive from genetic interactions and not from biochemical studies and this is not reflected in the title.

Our response: Thanks for the suggestion. We have emphasized our genetic studies and changed the title to “A trimeric CrRLK1L-LLG1 complex genetically modulates SUMM2-mediated autoimmunity”.

-In the Title: “A trimeric complex regulates...”; in the Abstract: “We have identified a trimeric complex..that senses the disturbance of immune signaling and regulates the

activation of NLR SUMM2 for initiating cell death and autoimmunity”.

No direct evidence is shown for a complex including LET1, LET2/MDS1, and LIG1, for its ability to sense disturbances in immune signaling and to regulate signaling pathways.

Our response: In the revised manuscript, we have provided additional data to show the complex formation between LET1 and LET2/MDS1 (Fig. 4L). In addition, we have revised this sentence to “Thus, our data suggest that a trimeric complex consisting of two CrRLK1Ls LET1, LET2/MDS1, and a GPI-anchored protein LIG1 that likely senses the disturbance of MEKK1-MKK1/2-MPK4 signaling and regulates the activation of NLR SUMM2 for initiating cell death and autoimmunity.”

Introduction:

“LET1 and LET2 hetero-dimerize..”

According to the experimental data shown, LET1 and LET2 coimmunoprecipitate (Fig. 4E) and are in close proximity as observed by FRET experiments (Fig. 4F-G). This is sufficient to support the notion that they interact either directly or indirectly. Whether they are part of a multiprotein complex, a dimer or a trimer it remains to be established.

Our response: We concur with this reviewer that co-immunoprecipitation and FRET-FLIM experiments provide evidence that LET1 and LET2/MDS1 are in the close proximity in the cells. We further performed a pull-down assay with purified LET2/MDS1 extracellular domain (LET2^{ex}) from *E. coli* to pull-down LET1-FLAG expressed in protoplasts. The new result (Fig. 4L) shows that LET2^{ex} could pull down LET1. The data strengthened our conclusion that LET1 and LET2/MDS1 are in a protein complex, and heteromerize. We agree with this reviewer that we do not know whether they form hetero-dimer, hetero-trimer, or hetero-tetramer. We thus changed “LET1 and LET2/MDS1 hetero-dimerize.” to “LET1 and LET2/MDS1 heteromerize” in the revision.

“LET2 promotes LET1 phosphorylation suggesting a phosphoregulation between different CrRLK1Ls”. The evidence for this claim is circumstantial and not supported by follow up analysis. Is LET2 a functional kinase? Does it phosphorylates LET1 *in vitro*? Is LET1 and LET2 kinase activity and/or phosphorylation modulated *in vivo* by activation of cell death (e.g. MEKK1 silencing)? The evidence that LET1 is phosphorylated when coexpressed with LET2 and that a mutation in the LET2 ATP binding site make LET2 non-functional is interesting, but it is not sufficient to provide an activation mechanism for LET1 by LET2/MDS1.

Our response: In the revised manuscript, we have shown that both LET1 and LET2/MDS1 are functional kinases by an *in vitro* kinase assay with purified proteins of cytosolic domains of LET1 and LET2/MDS1 (new Fig. 1I). We also showed that LET2/MDS1, not its kinase-inactive mutant, activated the kinase activity of LET1 *in vitro* when LET1 and LET2/MDS1 were coexpressed in protoplasts and immunoprecipitated for an *in vitro* kinase assay (new Fig. 4H). We have shown that the kinase-inactive mutants of LET2/MDS1 (Fig. 1H), and LET1 (Liu, Huang et al., submitted) could not complement their function in RNAi-*MEKK1* cell death.

Therefore, the LET1 and LET2/MDS1 kinase activity/phosphorylation *in vivo* is important for the activation of cell death. We concur with this reviewer that the detailed activation mechanism of LET1 by LET2/MDS1 still awaits to be elucidated.

“Thus, a specific trimeric module consisting of LET1, LET2 and GPI-anchored LLG1 senses the disturbance caused by a deficient MEKK1-MKK1/2-MPK4 cascade, and modulates SUMM2-mediated autoimmunity.” As mentioned above, there is no biochemical or other evidence that LET1, LET2 or LLG1 senses the effect of an inactive MEKK1-MKK1/2-MPK4 cascade. There is also no proof that they form a trimeric module and regulate SUMM2 activity.

Our response: We thank this reviewer for the comments, and have made the change in the revision “Thus, our results suggest that a specific trimeric *CrRLK1L* module consisting of LET1, LET2/MDS1 and the GPI-anchored LLG1 senses the disturbance caused by a deficient MEKK1-MKK1/2-MPK4 cascade, and modulates SUMM2-mediated autoimmunity.”.

Results:

-The rationale and minimal background information for the various experiments is often missing making hard to understand the experimental design and the hypotheses behind it.

Our response: Thanks for the suggestion. We have added additional rationale and background information in the results.

-Page 5 paragraph: “The mutations in..”

It is not clear at a first sight why LET1 is excluded from the targeted screen. It should be clearly mentioned that a mutation in LET1 was found in a previous screen to inhibit the cell death induced by MEKK1 silencing. This knowledge also provides the rationale to test other family members.

Our response: We have introduced LET1 in the last paragraph of Introduction. Here, we also added one more sentence to provide some background information about LET1 in Results “Among them, *LET1 (AT2G23200)* was identified as a regulator of autoimmunity in *mekk1*, *mkk1/2* and *mpk4* (Liu, Huang et al., manuscript submitted).”.

-Figure 1. LET2/MDS1 should be consistently indicated with both names in all parts of the manuscript (including Fig. 1D).

Our response: We thank this reviewer’s suggestion, and have indicated LET2/MDS1 in the revised manuscript.

-Page 6: “Thus the data support that only LET2/MDS1... is involved in the regulation of mekk1 cell death.” Here and in many other parts of the manuscript, the term "regulation" is inappropriately used. The genetic experiments presented here indicate that LET2/MDS1 plays a role in the activation of cell death caused by MEKK1 silencing. No regulatory role is demonstrated for LET2/MDS1.

Our response: We have taken this reviewer’s suggestion, and changed “regulation” to “modulation” or “regulate” to “modulate” in this place and other parts of the manuscript.

-Page 7: “The let2mekk1 mutant was slightly smaller than let1mekk1...”

It is the first time that this mutant is mentioned in the text. The comparison is not obvious and thereon why the let1mekk1 mutant is used should be explained.

Our response: We have taken this reviewer’s suggestion, and added a sentence “Since the *let1mekk1* double mutant also suppressed the growth defects of *mekk1* (Liu, Huang et al., manuscript submitted), we compared the phenotype of *let2mekk1* and *let1mekk1*.” in the revision.

-Page 8: “Since LET2/MDS1 functions in the same pathway with MEKK2 for SUMM2 activation...”

This claim at the present stage of the manuscript is not supported by results shown so far. Up to this point, the only evidence related to the possibility that LET2/MDS1 is in the same pathway of MEKK2 is the observation that the phenotype of the *mekk2let1/2mpk4* quadruple mutant is similar to that of the *let1/2mpk4* or *mekk2mpk4*. This is not sufficient to conclude that LET2/MDS1 is involved in SUMM2 activation through MEKK2.

Our response: We have changed this sentence to “Since both LET2/MDS1 and MEKK2 are required for SUMM2 activation, we tested the genetic relationship of LET2/MDS1 with MEKK2 and SUMM2”.

-Page 9: “Taken together, LET2/MDS1 functions genetically downstream of MEKK2 and upstream of SUMM2 in modulating the mekk1-mkk1/2-mpk4 cell death pathway”. Again, overinterpretation of the results: the evidence that overexpression of an active SUMM2 variant causes a similar phenotype in *let2* mutant and wt plants is not sufficient to claim that LET2/MDS1 functions upstream of SUMM2.

Our response: Overexpression of active SUMM2 caused a similar cell death in WT and *let2* mutant, suggesting that LET2/MDS1 is not required for active SUMM2-triggered cell death and might act independently or upstream of SUMM2. LET2/MDS1 is involved in the SUMM2-activated *mekk1-mkk1/2-mpk4* cell death. Thus, it is likely that LET2/MDS1 functions genetically upstream of SUMM2 in the *mekk1-mkk1/2-mpk4* cell death pathway. However, we cannot rule out the possibility that LET2/MDS1 functions independently of SUMM2 in *mekk1-*

mkk1/2-mpk4 cell death pathway. We have added this sentence and modified other sentences in the revision. Now it reads as “The data indicate that LET2/MDS1 is not required for active SUMM2-triggered cell death and might act independently or upstream of SUMM2. Taken together, our results suggest that LET2/MDS1 functions genetically downstream of MEKK2 and upstream of SUMM2 in the *mekk1-mkk1/2-mpk4* cell death pathway. However, we cannot rule out the possibility that LET2/MDS1 functions independently of SUMM2 in the *mkk1-mkk1/2-mpk4* cell death pathway.”

Also, it appears that the *mkk1-mkk1/2-mpk4* cell death pathway modulates LET2/MDS1 and MEKK2, and not the opposite.

Our response: To avoid confusion, we have deleted “modulating” in the revision. Now, it reads as “Taken together, our results suggest that LET2/MDS1 functions genetically downstream of MEKK2 and upstream of SUMM2 in the *mkk1-mkk1/2-mpk4* cell death pathway.”

-Fig. 4B: The data presented showing that LET2 accumulation is increased by coexpression of MEKK2 is not enough reliable: appropriate controls are missing in this experiment: 1. Coexpression of LET-HA and MEKK2-GFP should be compared to coexpression of LET-HA and GFP; 2. The Coomassie blue staining of Rubisco is not sufficient to demonstrate equal loading of samples, particularly in an experiment where protein accumulation is monitored. More importantly, in protoplasts there is no effect of MEKK2-FLAG expression on LET2-HA accumulation (Fig. 4A): in the input panels of the figure, there is no difference in the accumulation of LET2-HA in the presence or absence of MEKK2-FLAG, lanes 1 and 2.

Our response: We thank this reviewer’s suggestion, and have added additional data to strengthen our conclusion about LET2/MDS1 stabilization by MEKK2. We have included a control of GFP to show that GFP did not affect LET2/MDS1 protein accumulation in *Nicotiana benthamiana* (new Fig. 4B compare lane 1 and 2). In addition, we also included another control to show that MEKK2 did not affect GFP protein accumulation in *N. benthamiana* (new Fig. 4B compare lane 4 and 5). Furthermore, LET2/MDS1 proteins were stabilized by the treatment of MG132, a proteasome-dependent protein degradation inhibitor, in *35S::LET2-HA* transgenic plants, and in *N. benthamiana* (new Fig. 4C and D). Notably, the effect of MG132 is less pronounced in the presence of MEKK2, suggesting that MEKK2 had a similar effect with MG132 on the stabilization of LET2-HA (new Fig. 4D). The defect of MEKK1-MKK1/2-MPK4 pathway induced accumulation of MEKK2 transcripts and proteins (Su et al., 2013), which might lead to the stabilization of LET2/MDS1. Consistent with this hypothesis, the amount of LET2-HA proteins was increased in three independent *35S::LET2-HA* transgenic plants upon silencing *MEKK1* by VIGS (new Fig. 4E). Collectively, these results suggest that MEKK2 modulates LET2/MDS1 protein homeostasis.

When we performed Co-IP assay, we have adjusted the input amount to a similar level to compare Co-IP efficiency. In addition, we usually do not use protoplast transient assay to evaluate protein accumulation since the proteins are constantly expressed in protoplasts within the time frame we collect samples (within 12 hr). We thus performed multiple experiments in

stable transgenic plants and in *N. benthamiana* to investigate LET2/MDS1 protein accumulation (Fig 4B-E).

-Fig. 4C-D: The observation that LET1 is phosphorylated when coexpressed with LET2/MDS1 is intriguing, but not sufficiently followed up for drawing solid conclusions about the role of LET2/MDS1 kinase activity in signaling: is LET2/MDS1 an active protein kinase? Is LET1 phosphorylated *in vivo* upon induction of cell death by an inactive MEKK1-MKK1/2-MPK4 cascade or by overexpression of MEKK2? Is this phosphorylation required for LET1 activation?

Our response: Those are the similar questions that the reviewer asked in the Introduction. We briefly restated here. In the revised manuscript, we have shown that both LET1 and LET2/MDS1 are functional kinases by an *in vitro* kinase assay with purified proteins of cytosolic domains of LET1 and LET2/MDS1 (new Fig. 1I). We also showed that LET2/MDS1, not its kinase-inactive mutant, activated the kinase activity of LET1 *in vitro* with LET1 and LET2/MDS1 proteins immunoprecipitated from protoplasts and used in an *in vitro* kinase assay (new Fig. 4H). We have shown that the kinase-inactive mutants of LET2/MDS1 (Fig. 1H), and LET1 (Liu, Huang et al., submitted) could not complement their function in RNAi-*MEKK1* cell death. Therefore, the LET1 and LET2/MDS1 kinase activity/phosphorylation *in vivo* is important for the activation of cell death.

-Page 10 “We tested whether LRE/LLGs are involved in LET1/2-mediated *mekk1* cell death..”

How many LREs and LLGs are encoded in the Arabidopsis genome? How many were tested? How single and double mutants included in this study were selected?

Our response: We have clarified this in the revision. Now, the sentences read as “The GPI-anchored proteins LRE, LLG1, LLG2 and LLG3, function as adaptors/co-receptors for CrRLK1Ls FER and BUPs/ANXs. LLG2 and LLG3 function redundantly in regulating pollen tube integrity. We tested whether LRE/LLGs are involved in LET1/2-mediated *mekk1* cell death by silencing *MEKK1* in the corresponding single and double mutants, including two *lre* mutant alleles (*lre-3* and *lre-6*), two *llg1* mutant alleles (*llg1-1* and *llg1-2*), *llg2-1*, and *llg3-1* single mutants, and *llg2-1llg3-1* double mutant.”

-Page 11. The authors point out that their observations related to the *llg1* mutants are apparently contradicting: the *llg1* mutants aggravated the growth defects caused by a mutation in *mekk1*, whereas they suppressed cell death triggered by VIGS of the *MEKK1* gene. This contradiction is explained by the authors hypothesizing that “LGG1 plays one role in regulating initial seedling development in concert with *MEKK1*, and another role in regulating *mekk1* cell death at later stages” and that “compared to genetic mutations, VIGS-mediated silencing bypasses the defects associated with embryonic and early seedling development.” If this is indeed the case, why subsequent experiments were carried out with *mkk1/2* and *mpk4* genetic mutants and not by silencing these genes by VIGS? Does *MEKK1* signal through *MKK1/2* and *MPK4* during seedling development?

Our response: We thank this reviewer for pointing this out. We have tried to silence *MKK1/2* or *MPK4* in Col-0 by VIGS. However, the plants only showed weak cell death phenotype upon silencing *MKK1/2* or *MPK4*. Thus, it is difficult to evaluate the suppressors of *mkk1/2* and *mpk4* cell death with VIGS method, and we used *mkk1/2* and *mpk4* genetic mutants in the experiments. This could be due to less silencing efficiency of RNAi-*MKK1/2* and RNAi-*MPK4* than RNAi-*MEKK1*. Alternatively, unlike *MEKK1*, the remaining transcripts of *MKK1/2* or *MPK4* in the silenced plants might be sufficient for their functions. Notably, *mekk1* has stronger growth defects than *mpk4* and *mkk1/2* mutants (Qiu et al., 2008; Su et al., 2007), which might also explain why the growth defects in RNA-*MEKK1* plants are more obvious than RNAi-*MKK1/2* and RNAi-*MPK4* plants. The MEKK1 signaling in seedling development might not be mediated by MKK1/2 and MPK4 since the *llg1* mutants did not enhance the growth defects of *mkk1/2* and *mpk4* mutants (Fig. 6), which is different with *llg1mekk1* (Fig. 5). This conclusion is also supported by the observation that *mekk1* has stronger growth defects than *mpk4* and *mkk1/2* mutants, and *mekk1*, but not *mkk1/2* and *mpk4*, also has defects in the root length and branching patterns (Qiu et al., 2008; Su et al., 2007).

Discussion:

-Page 15: “In contrast, LET2/MDS1 share a similar function in regulating SUMM2 activation with LET1.” There is no evidence in this study for regulation of SUMM2 activation by LET2/MDS1 or LET.

Our response: We have deleted this sentence in the revision.

-Page 15: “We also show that LET2/MDS1 interacts with MEKK2 and SUMM2 and its stability is regulated by MEKK2”. As indicated above, the evidence for regulation of LET2/MDS1 stability by MEKK2 is very weak and does not support this conclusion.

Our response: As addressed previously, we have strengthened our data about LET2/MDS1 stabilization by MEKK2 in the revision. Please see our response to Fig. 4B for this reviewer.

-Page 15: “The data indicate that LET2/MDS1 may function upstream of LET1 and phosphorylate LET1 for signaling activation.” The data supporting this possibility are preliminary and limited to the observation that when the two proteins are transiently overexpressed in *N. benthamiana* plants LET1 is phosphorylated and this phosphorylation is only observed with LET2/MDS1 in the wild-type form but not with a putative kinase deficient mutant. This observation should be corroborated by other experiments before any conclusion can be drawn about the role of LET1 phosphorylation for signaling activation.

Our response: We have deleted this sentence in the revision.

-Page 15: “Consistent with this observation, mutations of either LET1 or LET2/MDS1 in the *let1* or *let2* single mutants disrupt LET1/LET2 complex and signaling pathway, leading to inactivation of SUMM2.” These data do not appear in the manuscript.

Our response: We have deleted “disrupt LET1/LET2 complex and signaling pathway”, and the new sentence now reads as “Consistent with this observation, mutations of either LET1 or LET2/MDS1 in the *let1* or *let2* single mutants lead to the inactivation of SUMM2 in *mekk1-mkk1/2-mpk4* cell death.”

-The manuscript Yang et al. (submitted), is cited in the text but not provided.

Our response: Yang et al manuscript is now online, and has been added.

Reference

- Kong, Q., Qu, N., Gao, M., Zhang, Z., Ding, X., Yang, F., Li, Y., Dong, O.X., Chen, S., Li, X., and Zhang, Y.** (2012). The MEKK1-MKK1/MKK2-MPK4 kinase cascade negatively regulates immunity mediated by a mitogen-activated protein kinase kinase in Arabidopsis. *The Plant cell* **24**, 2225-2236.
- Li, C., Yeh, F.L., Cheung, A.Y., Duan, Q., Kita, D., Liu, M.C., Maman, J., Luu, E.J., Wu, B.W., Gates, L., Jalal, M., Kwong, A., Carpenter, H., and Wu, H.M.** (2015). Glycosylphosphatidylinositol-anchored proteins as chaperones and co-receptors for FERONIA receptor kinase signaling in Arabidopsis. *eLife* **4**.
- Qiu, J.L., Zhou, L., Yun, B.W., Nielsen, H.B., Fiil, B.K., Petersen, K., MacKinlay, J., Loake, G.J., Mundy, J., and Morris, P.C.** (2008). Arabidopsis mitogen-activated protein kinase kinases MKK1 and MKK2 have overlapping functions in defense signaling mediated by MEKK1, MPK4, and MKS1. *Plant Physiology* **148**, 212-222.
- Shen, Q., Bourdais, G., Pan, H., Robatzek, S., and Tang, D.** (2017). Arabidopsis glycosylphosphatidylinositol-anchored protein LIG1 associates with and modulates FLS2 to regulate innate immunity. *Proceedings of the National Academy of Sciences of the United States of America* **114**, 5749-5754.
- Su, S.H., Suarez-Rodriguez, M.C., and Krysan, P.** (2007). Genetic interaction and phenotypic analysis of the Arabidopsis MAP kinase pathway mutations *mekk1* and *mpk4* suggests signaling pathway complexity. *Febs Letters* **581**, 3171-3177.
- Yong, Y., Liu, j., Yin, C., de Souza Vespoli, L., Ge, D., Huang, y., Feng, B., Xu, G., Ana Marcia, Dou, S., Criswell, C., Shan, L., Wang, X., and He, P.** (2020). RNAi-based screen reveals concerted functions of MEKK2 and CRCK3 in plant cell death regulation. *Plant Physiol* DOI:10.1104/pp.1119.01555.
- Xiao, Y., Stegmann, M., Han, Z., DeFalco, T.A., Parys, K., Xu, L., Belkhadir, Y., Zipfel, C., and Chai, J.** (2019). Mechanisms of RALF peptide perception by a heterotypic receptor complex. *Nature* **572**, 270-274.

REVIEWERS' COMMENTS:

Reviewer #1 (Remarks to the Author):

The authors have fairly addressed my comments.
The manuscript is also considerably improved in clarity.
I have no further comments.

Reviewer #2 (Remarks to the Author):

The authors have carefully addressed the comments, and the revised manuscript is much improved. I do not have any further suggestions.